# Crush Optimism with Pessimism:
# Structured Bandits Beyond Asymptotic Optimality

**Kwang-Sung Jun**
The University of Arizona
kjun@cs.arizona.edu

**Chicheng Zhang**
The University of Arizona
chichengz@cs.arizona.edu

## Abstract

In this paper, we study stochastic structured bandits for minimizing regret. The fact that the popular optimistic algorithms do not achieve the asymptotic instance-dependent regret optimality (asymptotic optimality for short) has recently allured researchers. On the other hand, it is known that one can achieve a bounded regret (i.e., does not grow indefinitely with $n$) in certain instances. Unfortunately, existing asymptotically optimal algorithms rely on forced sampling that introduces an $\omega(1)$ term w.r.t. the time horizon $n$ in their regret, failing to adapt to the "easiness" of the instance. In this paper, we focus on the finite hypothesis class and ask if one can achieve the asymptotic optimality while enjoying bounded regret whenever possible. We provide a positive answer by introducing a new algorithm called CRush Optimism with Pessimism (CROP) that eliminates optimistic hypotheses by pulling the informative arms indicated by a pessimistic hypothesis. Our finite-time analysis shows that CROP $(i)$ achieves a constant-factor asymptotic optimality and, thanks to the forced-exploration-free design, $(ii)$ adapts to bounded regret, and $(iii)$ its regret bound scales not with the number of arms $K$ but with an effective number of arms $K_\psi$ that we introduce. We also discuss a problem class where CROP can be exponentially better than existing algorithms in *nonasymptotic* regimes. Finally, we observe that even a clairvoyant oracle who plays according to the asymptotically optimal arm pull scheme may suffer a linear worst-case regret, indicating that it may not be the end of optimism.

## 1 Introduction

We consider the stochastic structured multi-armed bandit problem with a fixed arm set. In this problem, we are given a known structure that encodes how mean rewards of the arms are inter-dependent. Specifically, the learner is given a space of arms $\mathcal{A}$ and a space of hypotheses $\mathcal{F}$ where each $f \in \mathcal{F}$ maps each arm $a \in \mathcal{A}$ to its mean reward $f(a)$. Define $[n] := \{1, \ldots, n\}$. At each time step $t \in [n]$, the learner chooses an arm $a_t \in \mathcal{A}$ and observes a (stochastic) noisy version of its mean reward $f^*(a)$ where $f^* \in \mathcal{F}$ is the ground truth hypothesis determined before the game starts and not known to her. After $n$ time steps, the learner's performance is evaluated by her cumulative expected (pseudo-)regret:

$$\mathbb{E}\,\text{Reg}_n = \mathbb{E}\left[ n \cdot \max_{a \in \mathcal{A}} f^*(a) - \sum_{t=1}^{n} f^*(a_t) \right]. \tag{1}$$

Minimizing this regret poses a well-known challenge in balancing between exploration and exploitation; we refer to Lattimore and Szepesvári [21] for the backgrounds on bandits. We define our problem precisely in Section 2.

Linear bandits, a special case of structured bandits, have gained popularity over the last decade with exciting applications (e.g., news recommendation) [6, 11, 1, 22, 9]. While these algorithms use the

celebrated optimistic approaches to obtain near-optimal *worst-case* regret bounds (i.e., $\widetilde{O}(\sqrt{dn})$ where $\widetilde{O}$ hides logarithmic factors and $d$ is the dimensionality of the model), Lattimore and Szepesvári [20] have pointed out that their *instance-dependent* regret is often far from achieving the asymptotic instance-dependent optimality (hereafter, asymptotic optimality). This observation has spurred a flurry of research activities in asymptotically optimal algorithms for structured bandits and beyond, including OSSB [10], OAM [16] for linear bandits, and DEL [23] for reinforcement learning, although structured bandits and their optimality have been studied earlier in more general settings [3, 14].

The asymptotically optimal regret in structured bandits is of order $c(f) \cdot \ln(n)$ for instance $f \in \mathcal{F}$ where $c(f)$ is characterized by the optimization problem in (2). Its solution $\gamma \in [0, \infty)^K$ represents the optimal allocation of the arm pulls over $\mathcal{A}$, and some arms may receive zero arm pulls; we call those with nonzero arm pulls the *informative arms*. On the other hand, it is well-known that structured bandits can admit bounded regret [8, 19, 5, 25, 15, 27]; i.e., $\limsup_{n\to\infty} \mathbb{E}\,\mathrm{Reg}_n < \infty$. This is because the hypothesis space, which encodes the side information or constraints, can contain a hypothesis $f$ whose best arm alone is informative enough so that exploration is not needed, asymptotically.

However, existing asymptotically optimal strategies such as OSSB [10] cannot achieve bounded regret by design. The closest one we know is OAM [16] that can have a sub-logarithmic regret bound. The main culprit is their forced sampling, a widely-used mechanism for asymptotic optimality in structured bandits [10, 16]. Forced sampling, though details vary, ensures that we pull each arm proportional to an increasing but *unbounded* function of the time horizon $n$, which necessarily forces a *non-finite* regret. Furthermore, they tend to introduce the dependence on the number of arms $K$ in the regret unless a structure-specific sampling is performed, e.g., pulling a barycentric spanner in the linear structure [16].[1] While the dependence on $K$ disappears as $n \to \infty$, researchers have reported that the lower-order terms do matter in practice [16]. Such a dependence also goes against the well-known merit of exploiting the structure that their regret guarantees can have a mild dependence on the number of arms or may not scale with the number of arms at all (e.g., the worst-case regret of linear bandits mentioned above). We discuss more related work in the appendix (found in our supplementary material) due to space constraints, though important papers are discussed and cited throughout.[2]

Towards adapting to the easiness of the instance while achieving the asymptotic optimality, we turn to the simple case of the finite hypothesis space (i.e., $|\mathcal{F}| < \infty$) and ask: *can we design an algorithm with a constant-factor asymptotic optimality while adapting to finite regret?* Our main contribution is to answer the question above in the affirmative by designing a new algorithm and analyzing its finite-time regret. Departing from the forced sampling, we take a fundamentally different approach, which we call CROP (CRush Optimism with Pessimism). In a nutshell, at each time step $t$, CROP maintains a confidence set $\mathcal{F}_t \subseteq \mathcal{F}$ designed to capture the ground truth hypothesis $f^*$ and identifies two hypothesis sets: the optimistic set $\widetilde{\mathcal{F}}_t$ and the pessimistic set $\overline{\mathcal{F}}_t$ (defined in Algorithm 1). The key idea is to first pick carefully a $\overline{f}_t \in \overline{\mathcal{F}}_t$ that we call "pessimism", and then pull the *informative arms* indicated by $\overline{f}_t$. This, as we show, eliminates either the optimistic set $\widetilde{\mathcal{F}}_t$ or the pessimism $\overline{f}_t$ from the confidence set. Our analysis shows that repeating this process achieves the asymptotic optimality within a constant factor. Furthermore, our regret bound reduces to a finite quantity whenever the instance allows it and does not depend on the number of arms $K$ in general; rather it depends on an effective number of arms $K_\psi$ defined in (6). We elaborate more on CROP and the role of pessimism in Section 3. We present the main theoretical result in Section 4 and show a particular problem class where CROP's regret bound can be exponentially better than that of forced-sampling-based ones. Our regret bound of CROP includes an interesting $\ln\ln(n)$ term. In Section 5, we show a lower bound result indicating that such a $\ln\ln(n)$ term is unavoidable in general.

Finally, we conclude with discussions in Section 6 where we report a surprising finding that UCB can be in fact better than a clairvoyant oracle algorithm (that, of course, achieves the asymptotic optimality) in nonasymptotic regimes. We also show that such an oracle can suffer a linear worst-case regret under some families of problems including linear bandits, which we find to be disturbing, but this leaves numerous open problems.

## 2  Problem definition and preliminaries

In the structured multi-armed bandit problem, the learner is given a discrete arm space $\mathcal{A} = [K]$, and a finite hypothesis class $\mathcal{F} \subset (\mathcal{A} \to \mathbb{R})$ where we color definitions in blue, hereafter. There exists an unknown $f^* \in \mathcal{F}$ that is the ground truth mean reward function. Denote by $n$ the time horizon of the problem. For every $f \in \mathcal{F}$, denote by $a^*(f) = \arg\max_{a \in \mathcal{A}} f(a)$ and $\mu^*(f) = \max_{a \in \mathcal{A}} f(a)$ the arm and the mean reward *supported by* $f$, respectively. We remark that the focus of our paper is not computational complexity but the achievable regret bounds. For ease of exposition, we make the unique best arm assumption as follows:[3]

**Assumption 1** (Unique best arm). *For every $f \in \mathcal{F}$, there exists a unique best arm $a^*(f)$, i.e., $a^*(f)$ is singleton.*

For an arm $a$ and a hypothesis $f$, denote by $\Delta_a(f) = \mu^*(f) - f(a)$ the gap between the arm $a$ and the optimal arm, if the true reward function were $f$. Given a set of hypotheses $\mathcal{G}$, we denote by $a^*(\mathcal{G}) = \{a^*(f) : f \in \mathcal{G}\}$ and $\mu^*(\mathcal{G}) = \{\mu^*(f) : f \in \mathcal{G}\}$ the set of arms and mean rewards supported by $\mathcal{G}$ respectively.

The learning protocol is as follows: for each round $t \in [n]$, the learner pulls an arm $a_t \in \mathcal{A}$ and then receives a reward $r_t = f^*(a_t) + \xi_t$ where $\xi_t$ is an independent $\sigma^2$-sub-Gaussian random variable. The performance of the learner is measured by its expected cumulative regret over $n$ rounds defined in (1). Given an arm $a$ and time step $t$, denote by $T_a(t) = \sum_{s=1}^t \mathbb{1}\{a_s = a\}$ the arm pull count of $a$ up to round $t$. With this notation, $\mathbb{E}\operatorname{Reg}_n = \sum_{a \in \mathcal{A}} \mathbb{E}[T_a(n)]\Delta_a(f^*)$.

**Asymptotically optimal regret.**   Our aim is to achieve an asymptotic instance-dependent regret guarantee. Hereafter we abbreviate 'asymptotic optimality' to **AO**. Specifically, we would like to develop *uniformly good* algorithms, in that for any problem instance, the algorithm satisfies $\mathbb{E}\operatorname{Reg}_n = o(n^p)$ for any $p > 0$ where the little-o here is w.r.t. $n$ only. The regret lower bound of structured bandits is based on the *competing* class of functions $\mathcal{C}(f) = \{g : g(a^*(f)) = f(a^*(f)) \wedge a^*(g) \neq a^*(f)\}$. The class $\mathcal{C}(f)$ consists of hypotheses $g \in \mathcal{F}$ such that pulling arm $a^*(f)$ provides no statistical evidence to distinguish $g$ from $f$. Thus, even if the learner is confident that $f$ is the ground truth, she has to pull arms other than $a^*(f)$ to guard against the case where the true hypothesis is actually $g$ (in which case she suffers a linear regret); see the example in Figure 1(a) where $\mathcal{C}(f_4) = \{f_1\}$. The lower bound precisely captures such a requirement as constraints in the following optimization problem:

$$c(f) := \min_{\gamma \in [0,\infty)^K : \gamma_{a^*(f)}=0} \sum_a \gamma_a \Delta_a(f) \quad \text{s.t.} \quad \forall g \in \mathcal{C}(f), \ \sum_a \gamma_a \cdot \mathsf{KL}(f(a), g(a)) \geq 1 \ . \quad (2)$$

where $\mathsf{KL}(f(a), g(a))$ is the KL-divergence between the two reward distributions when the arm $a$ is pulled under $f$ and $g$ respectively. For the discussion of optimality, we focus on Gaussian rewards with variance $\sigma^2$, which means $\mathsf{KL}(f(a), g(a)) = \frac{(f(a)-g(a))^2}{2\sigma^2}$, though our proposed algorithm has a regret guarantee for more generic sub-Gaussian rewards. We denote by $\gamma(f)$ the solution of (2). Then, $c(f) = \sum_{a \in \mathcal{A}} \gamma_a(f) \cdot \Delta_a(f)$. The intuition is that if one could play arms in proportion to $\gamma^* = \{\gamma_a(f^*)\}_{a \in \mathcal{A}}$, then, by the constraints of the optimization problem, she would have enough statistical power to distinguish $f^*$ from all members of $\mathcal{C}(f^*)$; furthermore, $\gamma^*$ is the most cost-efficient arm allocation due to the objective function. The value of $\gamma^*$ can be viewed as the allocation that balances optimally between maximizing the *information gap* (i.e., the KL divergence in (2)) and minimizing the *reward gap* (i.e., $\Delta_a(f)$).

It is known from the celebrated works of Agrawal et al. [3] and Graves and Lai [14] that any uniformly good algorithm must have regret at least $(1 - o(1))c(f)\ln(n)$ for large enough $n$, under

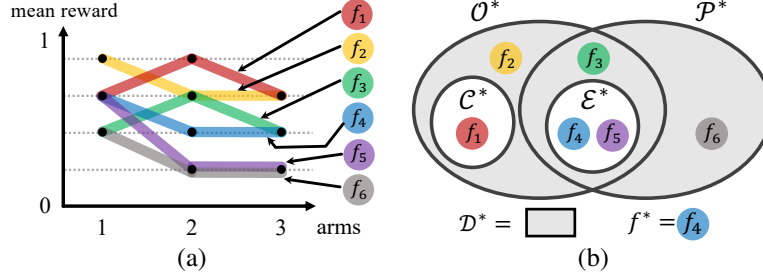

Figure 1: (a) An example instance. (b) A diagram of various hypothesis classes w.r.t. the ground truth hypothesis $f^*$. Best viewed in colors.

environment with ground truth reward function $f^* = f$. In other words, if an algorithm has a regret of $(1 - \Omega(1))c(f)\ln n$ under the ground truth $f$, then for large enough $n$, its expected arm pull scheme $\gamma = \left(\frac{\mathbb{E}[T_a(n)]}{\ln n}\right)_{a\in\mathcal{A}}$ must violate the constraint in (2) for some $g \in \mathcal{C}(f)$, implying that the algorithm must not be a uniformly good algorithm (i.e., suffer a polynomial regret under $g$). They also show the lower bound is tight by developing algorithms with asymptotic regret bound of $(1 + o(1))c(f)\ln n$.

**The oracle.** The lower bound suggests that one should strive to ensure $\mathbb{E}[T_a(n)] \approx \gamma_a^* \ln(n)$. Indeed a clairvoyant oracle (the oracle, hereafter) who knows $f^*$ would, at round $t$, pull the arm $a$ such that $T_a(t-1) \le \gamma_a^* \ln(t)$ if there exists such an arm (i.e., exploration), and otherwise pull the best arm (i.e., exploitation). The oracle will initially pull the informative arms only, but as $t$ increases, exploitation will crowd out exploration. We believe mimicking the oracle is what most algorithms with AO are after. Particularly, the most common strategy is to replace $\gamma^*$ with the Empirical Risk Minimizer (ERM) $\gamma(\widehat{f}_t)$ where $\widehat{f}_t \in \mathcal{F}$ is the one that best fits the observed rewards. Unlike supervised learning, however, the observed rewards are controlled by the algorithm itself, making the ERM brittle; i.e., the ERM may not converge to $f^*$. Thus, most studies employ a form of forced sampling to ensure that $\widehat{f}_t$ converges to $f^*$ so that $\gamma(\widehat{f}_t)$ converges to $\gamma^*$. As discussed before, this is precisely where the issues begin, and we will see that CROP avoids forced sampling and $\gamma(\widehat{f}_t)$ altogether.

**Example: cheating code.** We describe an example inspired by Amin et al. [4] when algorithms with AO provide an improvement over the popular optimistic algorithms. Let $K_0 \in \mathbb{N}_+$ and $e_i$ be the $i$-th indicator vector. The idea is to first consider a hypothesis like $f = (1, 1-\epsilon, 1-\epsilon, \ldots, 1-\epsilon)$ and then add those hypotheses that copy $f$, pick one of its non-best arms, and replace its mean reward with $1+\epsilon$. This results in total $K_0 - 1$ competing hypotheses. Specifically, let $e_i$ be the $i$-th indicator vector and define $h(i,j) \in \mathbb{R}^{K_0}$ as follows: $\forall i \in [K_0], h(i,0) = (1-\epsilon)\mathbf{1} + \epsilon e_i$ and $\forall j \in [K_0]\setminus\{i\}$, $[h(i,j)]_k = \begin{cases} 1+\epsilon & \text{if } k = j \\ h(i,0) & \text{otherwise} \end{cases}$. Let $\mathcal{F}_0 = \{h(i,j) : i \in [K_0], j \in \{0,1,\ldots,K_0\}\setminus\{i\}\}$, $k = \lceil\log_2(K_0)\rceil$, and $\Lambda \in [0, 1/2]$. Finally, we define the "cheating code" class with $K = K_0 + k$ arms:

$$\mathcal{F}^{\text{code}} = \left\{(g_{1:K_0}, \Lambda \cdot b_{1:k}) \in \mathbb{R}^{K_0+k} : g \in \mathcal{F}_0, b \in \{0,1\}^k\colon \text{binary representation of } a^*(g) - 1\right\},$$

which appends $k$ "cheating arms" that tells us the index of the best arm. Let us fix $f^* \in \mathcal{F}^{\text{code}}$ such that $\mu^*(f^*) = 1$. Assume $\frac{1}{2\epsilon} > \frac{2}{\Lambda^2}$ so that the informative arms of $f^*$ are the cheating arms (see the appendix for reasoning) where we color in green for emphasis, throughout. Let $\sigma^2 = 1$. For the instance $f^*$, an algorithm with a constant-factor AO would have regret $O(\frac{\log_2 K}{\Lambda^2}\ln(n))$ (elaborated more in the appendix). In contrast, optimistic algorithms such as UCB [7] (i.e., run naively without using the structure) or UCB-S [19], would pull the arm $\widetilde{a}_t$ where

$$(\widetilde{a}_t, \widetilde{f}_t) = \arg\max_{a\in\mathcal{A}, f\in\mathcal{F}_t} f(a) \tag{3}$$

and $\mathcal{F}_t$ is a confidence set designed to trap $f^*$ with high probability. One can show that $\widetilde{a}_t$ is always one of the first $K_0$ arms and that their regret is $O(\frac{K}{\epsilon}\ln(n))$, which can be much larger. In fact, the gap between the two bounds can be arbitrarily large as $\epsilon$ approaches to 0.

**The anatomy of the function classes.** There are function classes besides $\mathcal{C}(f)$ that will become useful in our study. We first define an equivalence relationship between hypotheses: we call $f \sim g$

---
**Algorithm 1** CRush Optimism with Pessimism (CROP)
---
**Require:** The hypothesis class $\mathcal{F}$, parameters $z, \mathring{z}, \alpha, \mathring{\alpha} > 1$

1: **for** $t = 1, 2, \ldots, n$ **do**
2:      Let $\mathcal{F}_t = \{f \in \mathcal{F} : L_{t-1}(f) - \min_{g \in \mathcal{F}} L_{t-1}(g) \le \beta_t := 2\sigma^2 \ln\left(zt(\log_2 t)^2\right)\}$
3:      **if** $a^*(\mathcal{F}_t)$ is singleton **then**
4:         (`Exploit`) Pull the arm $a_t \in a^*(\mathcal{F}_t)$, observe the reward $r_t$.
5:         Continue to the next iteration.
6:      **end if**
7:      Let $\mathcal{B}_t = \{(a^*(f), \mu^*(f)) : f \in \mathcal{F}_t\}$ be the **b**est arm candidate set.
8:      Find the optimistic arm, mean, and set:
$$(\widetilde{a}_t, \widetilde{\mu}_t) = \arg\max_{(a,\mu) \in \mathcal{B}_t} \mu, \quad \widetilde{\mathcal{F}}_t = \mathcal{F}_t(\widetilde{a}_t, \widetilde{\mu}_t) \ .$$
9:      Find the pessimistic arm, mean, set, and hypothesis:
$$(\overline{a}_t, \overline{\mu}_t) = \arg\min_{(a,\mu) \in \mathcal{B}_t : \, a \ne \widetilde{a}_t} \mu, \quad \overline{\mathcal{F}}_t = \mathcal{F}_t(\overline{a}_t, \overline{\mu}_t), \quad \overline{f}_t = \arg\min_{f \in \overline{\mathcal{F}}_t} L_{t-1}(f) \ .$$
10:    Define $\mathring{\mathcal{F}}_t = \{f \in \overline{\mathcal{F}}_t : L_{t-1}(f) - L_{t-1}(\overline{f}_t) \le \mathring{\beta}_t := 2\sigma^2 \ln(\mathring{z}(\log_2(t))^{\mathring{\alpha}})\}$.       (let $\mathring{\beta}_1 = \infty$)
11:    **if** $\exists f, g \in \mathring{\mathcal{F}}_t$ s.t. $\gamma(f) \not\sim \gamma(g)$ **then**
12:       (`Conflict`) $\pi_t = \phi(\overline{f}_t)$ .                                                          (see (5))
13:    **else if** $\gamma(\overline{f}_t)$ satisfies that $\forall f \in \widetilde{\mathcal{F}}_t, \sum_a \gamma_a(\overline{f}_t) \frac{(\overline{f}_t(a) - f(a))^2}{2\sigma^2} \ge 1$, **then**
14:       (`Feasible`) $\pi_t = \gamma(\overline{f}_t)$.
15:    **else**
16:       (`Fallback`) $\pi_t = \psi(\overline{f}_t)$.                                                               (see (4))
17:    **end if**
18:    Pull arm $a_t = \arg\min_a \frac{T_a(t-1)}{\pi_{t,a}}$    (take $\frac{x}{0}$ with $x \ge 0$ as $\infty$; break ties arbitrarily), and then observe the reward $r_t$.
19: **end for**
---

if $a^*(f) = a^*(g)$ and $\mu^*(f) = \mu^*(g)$; one can verify that it satisfies reflexiveness, symmetry, and transitivity, and induces a partition over $\mathcal{F}$. Given $f \in \mathcal{F}$, we denote by $\mathcal{E}(f)$ the equivalent class $f$ belongs to and by $\mathcal{D}(f) = \{g : g(a^*(f)) \ne f(a^*(f))\}$ its *docile* class that can be easily distinguished from $f$ as we describe later. One can show that for every $f \in \mathcal{F}$, the class $\mathcal{F}$ is a disjoint union of $\mathcal{E}(f)$, $\mathcal{D}(f)$, and $\mathcal{C}(f)$. We also define $\mathcal{O}(f) = \{g : \mu^*(g) \ge \mu^*(f)\}$ (and $\mathcal{P}(f) = \{g : \mu^*(g) \le \mu^*(f)\}$) as the set of hypotheses that support mean rewards that are not lower (and not higher) than $\mu^*(f)$ (respectively). We use shorthands $\mathcal{E}^* := \mathcal{E}(f^*)$ and $\mathcal{D}^*, \mathcal{C}^*, \mathcal{O}^*$, and $\mathcal{P}^*$ defined similarly. We draw a Venn diagram of theses classes in Figure 1(b) along with the example hypotheses in Figure 1(a); we recommend that the readers verify the example themselves to get familiar with these classes.

**Bounded regret.** When the ground truth $f^*$ enjoys $\mathcal{C}^* = \varnothing$, then $c(f^*) = 0$ and the algorithm can achieve bounded regret, which is well-known as mentioned in our introduction. This is because, when $f^*$ has no competing hypothesis, pulling the best arm $a^*(f^*)$ alone provides a nonzero statistical evidence that distinguishes $f^*$ from $\mathcal{F} \setminus \mathcal{E}^* = \mathcal{D}^*$. That is, there is no need to explore as exploitation alone provides sufficient exploration.

## 3   Crush Optimism with Pessimism (CROP)

We now introduce our algorithm CROP. First, some definitions: for any $\mathcal{G}$, define $\mathcal{G}(a, \mu) = \{f \in \mathcal{G} : a^*(f) = a, \mu^*(f) = \mu\}$. Given a set of observations $\{(a_s, r_s) : s \in [t]\}$ up to time step $t$, and $f \in \mathcal{F}$, denote by $L_t(f) = \sum_{s=1}^t (f(a_s) - r_s)^2$ the cumulative squared loss of $f$ up to time step $t$. We use this loss to construct a confidence set that captures the ground truth $f^*$, which is inspired by Agarwal et al. [2], but we extend theirs to allow sub-Gaussian rewards. The loss $L_t(f)$ gives a measure of goodness of fit of hypothesis class $f$, in that $f^*$ is the Bayes optimal regressor that minimizes $\mathbb{E} L_t(f)$.

We describe CROP in Algorithm 1, where the parameters $\{\alpha, \mathring{\alpha}\}$ are numerical constants and $\{z, \mathring{z}\}$ should be set to $|\mathcal{F}|$ (precise defined in Theorem 1). CROP has four main branches: `Exploit`, `Conflict`, `Feasible`, and `Fallback`. Note that `Feasible` is the main insight of the algorithm that we focus first while `Conflict` deals with some difficult cases, which we describe the last.

**Exploit.** At every round $t$, CROP maintains a confidence set $\mathcal{F}_t$, the set of hypotheses $f$ in $\mathcal{F}$ that fits well with the data observed so far w.r.t. $L_{t-1}(f)$. This is designed so that the probability of failing to trap the ground truth hypothesis $f^*$ is $O\left(\frac{1}{t^\alpha}\right)$. We first check if $a^*(\mathcal{F}_t)$ is a singleton. If true, we pull the arm $a^*(\mathcal{F}_t)$ that is unanimously supported by all $f$ in $\mathcal{F}_t$. Note that the equivalence relationship $\sim$ induces a partition of $\mathcal{F}_t$. If we do not enter the exploit case, we select the equivalence class $\widetilde{\mathcal{F}}_t$ that maximizes its shared supported mean reward; we call this the *optimistic set*. This is related to the celebrated "optimism in the face of uncertainty" (OFU) principle that pulls arm $a_t$ by (3). In line 9, we deviate from the OFU and define the *pessimistic set* $\overline{\mathcal{F}}_t$, which is the equivalence class in $\mathcal{F}_t$ that *minimizes* its shared supported mean reward $\mu^*(\overline{\mathcal{F}}_t)$ with a constraint that they support an action other than $\widetilde{a}_t$. We then define $\overline{f}_t$, which we call *the pessimism*, as the Empirical Risk Minimizer (ERM) over $\overline{\mathcal{F}}_t$. Next, we compute $\mathring{\mathcal{F}}_t$, a refined confidence set inside the pessimistic set $\overline{\mathcal{F}}_t$, and then test a condition to enter `Conflict`; we will discuss it later as mentioned above. For now, suppose that we did not enter `Conflict` and are ready to test the condition for `Feasible` (line 13).

**Feasible.** The condition in line 13 first computes $\gamma(\overline{f}_t)$ and then tests whether all the hypotheses in $\widetilde{\mathcal{F}}_t$ satisfy the information constraint that takes the same form as those in the optimization problem for $c(\overline{f}_t)$. If this is true, then we set $\pi_t = \gamma(\overline{f}_t)$ and then move onto line 18 to choose which arm to pull. The intention here is to pull the arm that is most far away from the pull scheme of $\gamma(\overline{f}_t)$, which is often referred to as "tracking" [13]. Note that the arm $\overline{a}_t$ is never pulled because $\gamma_{\overline{a}_t}(\overline{f}_t) = 0$.

**Why the pessimism?** To motivate the design choice of tracking the pessimism, consider the example hypothesis space $\mathcal{H}$ in Figure 2. Suppose that at time $t$ we have $\mathcal{F}_t = \mathcal{H} = \{f_1, f_2, f_3\}$. Which arms should we pull? The OFU tells us to pull the optimistic arm $\widetilde{a}_t$ as done in Lattimore and Munos [19], but it does not achieve the instance optimality. Another idea mentioned in Section 2 is find the ERM $\widehat{f}_t = \arg\min_{f \in \mathcal{F}} L_{t-1}(t)$ and then pull the arms by tracking $\gamma(\widehat{f}_t)$; i.e., $a_t = \arg\min_{a \neq a^*(\widehat{f}_t)} T_a(t - 1)/\gamma_a(\widehat{f}_t)$. This is essentially the main idea of OSSB [10].[4] OAM [16] also relies on the ERM $\widehat{f}_t$, though they partly use the optimism. However, ERMs are brittle in bandits.

| Arms | A1 | A2 | A3 | A4 | A5 |
|------|------|------|------|------|------|
| $f_1$ | **1** | .99 | .98 | 0 | 0 |
| $f_2$ | .98 | **.99** | .98 | .25 | 0 |
| $f_3$ | .97 | .97 | **.98** | .25 | .25 |
| $f_4$ (optional) | .98 | **.99** | .98 | .25 | .50 |

Figure 2: The "staircase" example. Define $\mathcal{H} = \{f_1, f_2, f_3\}$ and $\mathcal{H}^+ = \{f_1, f_2, f_3, f_4\}$. We boldface the best arm and underline the informative arms of each hypothesis.

For example, when $f^* = f_3$, in earlier rounds, the ERM $\widehat{f}_t$ can be $f_2$ with nontrivial probability. Pulling the informative arm of $f_2$, which is A4, eliminates $f_1$ but will not eliminate $f_3$, and we get stuck at pulling A4 indefinitely. To avoid such a trap, researchers have introduced forced sampling.

What are the robust alternatives to the ERM? For now, suppose that $f^*$ is always in the confidence set $\mathcal{F}_t$. Among $\{\gamma(f_1), \gamma(f_2), \gamma(f_3)\}$, which one should we track? We claim that we should follow the pessimism, which is $f_3$ in this case. Specifically, if $f^* = f_3$, we are lucky and following the pessimism will soon remove both $f_1$ and $f_2$ from $\mathcal{F}_t$. We then keep entering `Exploit` and pull the best arm A3 for a while. Note that $f_1$ or $f_2$ will come back to $\mathcal{F}_t$ again as pulling A3 provides the same loss to every $f \in \mathcal{H}$ but the threshold $\beta_t$ of the confidence set $\mathcal{F}_t$ increases over time. In this case, the principle of pessimism will do the right thing, again.

What if $f^*$ was actually $f_2$? Following the pessimism $f_3$ is not optimal, but it does eliminate $f_3$ from $\mathcal{F}_t$ because $f_2$ appears in the constraint of the optimization (2); after the elimination, we have $\mathcal{F}_t = \{f_1, f_2\}$ and $\overline{f}_t = f_2$, so the pessimism is back in charge. In sum, the key observation is that the optimal pull scheme $\gamma(f)$ is designed to differentiate $f$ from its *competing hypotheses* that support arms with higher mean rewards than that of $f$. Assuming the confidence set works properly, tracking the pessimism either does the right thing or, if $\overline{f}_t$ is not the ground truth, removes $\overline{f}_t$ from the confidence set (also the right thing to do). However, to make it work beyond this example, we need other mechanisms: `Fallback` and `Conflict`.

**Fallback.** When the condition of `Feasible` is not satisfied, we know that the arm pull scheme $\gamma(\overline{f}_t)$ will not be sufficient to remove every $f \in \widetilde{\mathcal{F}}_t$ – or, it could even be impossible. Thus, we should not track $\gamma(\overline{f}_t)$. Instead, we design a different arm pull scheme $\psi(f)$ defined below so that tracking $\psi(\overline{f}_t)$ can remove all members of $\widetilde{\mathcal{F}}_t$ in a cost-efficient manner. With the notation $\Delta_{\min}(f) = \min_{a \neq a^*(f)} \Delta_{a^*(f)}(f)$,

$$\psi(f) := \underset{\gamma \in [0,\infty)^K}{\arg\min} \quad \Delta_{\min}(f) \cdot \gamma_{a^*(f)} + \sum_{a \neq a^*(f)} \Delta_a(f) \cdot \gamma_a$$

$$\text{s.t.} \quad \forall g \in \mathcal{O}(f) \setminus \mathcal{E}(f): \quad \sum_a \gamma_a \frac{(f(a) - g(a))^2}{2\sigma^2} \geq 1 \tag{4}$$

$$\gamma \geq \phi(f) \vee \gamma(f)$$

where $\phi$ is defined in (5) and explained below and $x \succeq y$ means $x_a \geq y_a, \forall a$. The constraints above now ensure that $\psi(\overline{f}_t)$ provides a sufficient arm pull scheme to eliminate $\widetilde{\mathcal{F}}_t$ even if the condition of `Feasible` is not satisfied. Another difference from $\gamma(f)$ is that $\gamma_{a^*(f)}$ can be nonzero, but we use $\Delta_{\min}(f)$ instead of $\Delta_{a^*(f)}(f) = 0$ to avoid $\gamma_{a^*(f)} = \infty$. That said, there are other design choices for $\psi(f)$, especially given that $\psi$ appears only with the finite terms in the regret bound. We discuss more on the motivation and alternative designs of (4) in the appendix.

**Conflict.** This is an interesting case where the learner faces the challenge not in finding which arm is the best arm, but rather which *informative* arms and their pull scheme one should track. Specifically, consider the other example of $\mathcal{H}^+$ in Figure 2. Suppose at time $t$ we have $\mathcal{F}_t = \{f_1, f_2, f_4\}$ and the ground truth is $f_4$, which means $\mathcal{E}^* = \{f_2, f_4\}$. If CROP does not have the `Conflict` mechanism, it will use $\overline{f}_t$, the ERM among $\overline{\mathcal{F}}_t$, which can be either $f_2$ or $f_4$. However, as explained before, ERMs are brittle; one can see that it can get stuck at tracking $f_2$ with nontrivial probability and pull less informative arms. Interestingly, this would not incur a linear regret. Rather, the regret would still be like $\ln(n)$ but with a suboptimal constant of $c(f_2)$ rather than $c(f_4)$; one can adjust our example to make this gap $c(f_2) - c(f_4)$ arbitrarily large, making it arbitrarily far from the AO. On the other hand, a closer look at $f_2$ and $f_4$ reveals that A5 is the only arm that can help distinguish $f_2$ from $f_4$. One might attempt to change CROP so that it pulls A5 in such a case, which results in either removing $f_4$ from the confidence set if $f^* = f_2$ or removing $f_2$ if $f^* = f_4$. However, if $f^* = f_2$, this would introduce an extra $\ln(n)$ term in the regret bound since A5 is a noninformative arm, which again can lead to a suboptimal regret bound.

CROP resolves this issue by constructing a refined confidence set $\mathring{\mathcal{F}}_t$ with a more aggressive failure rate of $1/\ln(t)$ rather than the usual $1/t$, and use this confidence set to weed out conflicting pull schemes. If the refined set $\mathring{\mathcal{F}}_t$ still contains hypotheses with conflicting pull schemes, then CROP enters `Conflict` and computes a different allocation scheme:

$$\phi(f) = \underset{\gamma \in [0,\infty)^K : \gamma_{a^*(f)} = 0}{\arg\min} \quad \sum_a \Delta_a(f) \cdot \gamma_a$$

$$\text{s.t.} \ \forall g \in \mathcal{E}(f) : \gamma(g) \not\propto \gamma(f), \quad \sum_a \gamma_a \frac{(f(a) - g(a))^2}{2\sigma^2} \geq 1 \tag{5}$$

where we use the convention $0 \propto 0$. Consider $\mathcal{H}^+$ in Figure 2 with $\sigma^2 = 1$. Then, $\phi(f_2) = \phi(f_4) = (0, 0, 0, 0, \frac{2}{(.5)^2} = 8)$. Our regret analysis will show that the quantity $\phi(f)$ appears in the regret bound with $\ln(\ln(n))$ term only instead of $\ln(n)$, allowing us to achieve the AO within constant-factor.

## 4 Analysis

Before presenting our analysis, we define the effective number of arms $K_\psi$ as the size of the union of the supports of $\psi(f)$ for all $f \in \mathcal{F}$:

$$K_\psi = \left| \left\{ a : \exists f \in \mathcal{F}, \psi_a(f) \neq 0 \right\} \right| . \tag{6}$$

Define $\phi(\mathcal{G}) = \left( \max_{f \in \mathcal{G}} \phi_a(f) \right)_{a \in \mathcal{A}}$ and $\psi(\mathcal{G})$ similarly. Let $\Lambda_{\min} = \min_{f \in \mathcal{D}^*} \frac{|f(a^*) - f^*(a^*)|}{\sigma}$ the smallest information gap where $a^* := a^*(f^*)$. We use the shorthand $\Delta_{\max} := \max_a \Delta_a(f^*)$. We present our main theorem on the regret bound of CROP.

**Theorem 1.** *Let* $(\alpha, \mathring{\alpha}, z, \mathring{z}) = (2, 3, |\mathcal{F}|, |\mathcal{F}|)$*. Suppose we run CROP with hypothesis class $\mathcal{F}$ with the environment $f^* \in \mathcal{F}$. Then, CROP has the following anytime regret guarantee: $\forall n \geq 2$,*

$$\mathbb{E} \operatorname{Reg}_n \leq c_1 \cdot \left( P_1 \ln(n) + P_2 \ln(\ln(n)) + P_3 \left( \ln(|\mathcal{F}|) + \ln(Q_1) \right) + K_\psi \Delta_{\max} \right),$$

*where $c_1$ is a numerical constant, and*

$$P_1 = \sum_a \Delta_a \gamma_a^*, \quad P_2 = \sum_a \Delta_a \phi_a(\mathcal{E}^*), \quad P_3 = \sum_a \Delta_a \psi_a(\mathcal{F}), \quad and \quad Q_1 = \Lambda_{\min}^{-2} + K_\psi \left( 1 + \max_i \psi_i(\mathcal{F}) \right).$$

*Furthermore, when $\gamma^* = 0$, we have $P_1 = P_2 = 0$, achieving a bounded regret.*

*Proof.* The main proof is deferred to the appendix. One technical challenge is to deal with `Conflict` in CROP via our refined confidence set $\mathring{\mathcal{F}}_t$. The failure rate of $\mathring{\mathcal{F}}_t$ is set $\operatorname{poly}(1/\ln(t))$ rather than the usual $\operatorname{poly}(1/t)$.[5] For example, there is an event where $\mathring{\mathcal{F}}_t$ fails to capture $f^*$ but $f^*$ is still in $\mathcal{F}_t$, which would lead to a $\ln(n)$ regret; we manage to prove that this scenario contributes to an $O(1)$ term *in expectation* by showing that it happens with probability like $1/\ln(n)$ times (roughly speaking) using a technique that we call "regret peeling". To bound other $O(1)$ terms that are attributed to the docile class $\mathcal{D}^*$, we borrow techniques from Lattimore and Munos [19]. $\qquad\square$

Our main theorem provide a sharp non-asymptotic instance-dependent regret guarantee. The leading term $O\left( \sum_a \Delta_a \gamma_a^* \log(n) \right)$ implies that we achieve the AO up to a constant factor. The second term is of order $\ln\ln(n)$, which comes from our analysis on `Conflict`. The remaining terms are $O(1)$, which depends on properties of $\psi_a(\mathcal{F})$ and $\Lambda_{\min}$. Unlike many strategies that perform forced exploration on all arms [16, 10] to achieve the asymptotic optimality, our bound has no dependency on the number of arms $K$ at all, even in the finite terms, but rather depends on the effective number of arms $K_\psi$.

Note that $K$-free regret bounds still happens with optimistic algorithms; e.g., in $\mathcal{F}^{\text{code}}$ (defined in Section 2), UCB depends on $K_0$ rather than $K$, and one can add arbitrarily many cheating arms to make $K \gg K_0$. Bounded regrets also have been shown via optimism [19, 15, 27], but they are far from the AO in general. Our novelty is to obtain instance optimality, remove the dependency on $K$, and achieve bounded regret whenever possible, simultaneously. We make more remarks on $\ln\ln(n)$ term and how one can get rid of $\ln(|\mathcal{F}|)$ and handle infinite hypothesis spaces in the appendix.

**Example: Cheating code.** Let $\mathcal{F} = \mathcal{F}^{\text{code}}$ and $\sigma^2 = 1$, and fix $f^* \in \mathcal{F}$ such that $\mu^*(f^*) = 1$. Assume $\frac{1}{2\epsilon} > \frac{2}{\Lambda^2}$. Then, one can show that $\gamma^* = \psi(\mathcal{F}) = (0, \ldots, 0, \frac{2}{\Lambda^2}, \ldots, \frac{2}{\Lambda^2})$, where the first $K_0$ coordinates are zeros, and $\phi(\mathcal{E}^*) = 0$. We also have $|\mathcal{F}| = K_0^2$, $K_\psi = \lceil \log_2(K_0) \rceil$, and $\Lambda_{\min} = \epsilon$. Then, using $K_0 \leq K$,

$$\mathbb{E} \operatorname{Reg}_n = O\left( \frac{\ln(K)}{\Lambda^2} \ln\left( Kn \cdot \left( \frac{1}{\epsilon} + \frac{\ln(K)}{\Lambda^2} \right) \right) + \ln(K) \right),$$

which is $\approx \frac{\ln^2(K)}{\Lambda^2} \ln(\frac{n}{\epsilon})$ when taking the highest-order factors for each $(n, K, \epsilon, \Lambda)$. We speculate that $\ln(1/\epsilon)$ can be removed with a tighter analysis. We compare CROP to algorithms with AO that use forced sampling (FS in short). Say, during $n$ rounds, FS pulls every arm $\ln\ln(n)$ times each, introducing a term $O(\epsilon K \ln\ln(n))$ in the regret, but let us ignore the $\ln\ln(\cdot)$ factor. For FS, the best regret bound one can hope for is $O(K\epsilon + \frac{\log_2(K)}{\Lambda^2} \ln(n))$. To satisfy the condition $\frac{1}{2\epsilon} > \frac{2}{\Lambda^2}$, set $\Lambda = 1/2$ and $\epsilon = 1/32$. Then, CROP's regret is $O(\ln^2(K) \ln(n))$ whereas FS's regret is $O(K + \ln(K) \ln(n))$. When $K \approx n$, FS has a linear regret whereas CROP has $\ln^3(n)$ regret. If $K = 2^d$ for some $d$, then the gap between the two becomes more dramatic: $O(2^d + d\ln(n))$ of FS vs $O(d^2 \ln(n))$ of CROP, an exponential gap in the nonasymptotic regime.

# 5 Lower bound: Necessity of the $\Omega(\ln\ln(n))$ term

One may wonder if the $\ln(\ln(n))$ term in our upper bound is necessary to achieve the asymptotic optimality up to constant factors. We show that there exist cases where such a dependence is indeed required. In fact, our lower bound statement is stronger; in a hypothesis class we construct for lower

bound, even polynomial-regret algorithms must also pull a non-informative arm at least $\ln \ln(n)$ times.

**Theorem 2** (Informal version). *Assume the Gaussian noise model with $\sigma^2 = 1$. There exists a hypothesis class $\mathcal{F}$ and an absolute constant $C$ that satisfies the following: If an algorithm has $\mathbb{E} \operatorname{Reg}_n \leq O(n^u)$ for some $u \in [0,1)$ under an instance $f \in \mathcal{F}$, then there exists another instance $f' \in \mathcal{F}$ and an arm $i$ with $\gamma_i(f') = 0$ (i.e., non-informative arm) such that*

$$\mathbb{E}_{f'} T_i(n) = \Omega(\ln(1 + (1 - u)\ln(n)))$$

*where $\mathbb{E}_{f'}$ is the expectation under the instance $f'$.*

The constructed instance for the lower bound is a variation of $\mathcal{H}^+$ in Figure 2. Our theorem shows that, just because an arm is noninformative, it does not mean that we can pull it a finite number of times.

Our result also has an implication for algorithms with forced sampling. To be specific, suppose that an algorithm $A$ performs forced sampling by requiring each arm to be pulled at least $\tau$ where $\tau$ is fixed at the beginning. Then, to achieve sublinear regret bounds, it is required that $A$ use $\tau = \omega(\ln(\ln(n))$. We emphasize that even $\tau = \Theta(\ln(\ln(n)))$ can suffer a polynomial regret, let alone being uniformly good. This is because the constant factor matters and is a function of the problem. We provide the precise constants, the full statement of the theorem, and its proof in our appendix.

## 6 Discussion

There are improvements to be made including more examples and studying properties of alternative designs of $\phi$ and $\psi$, which we discuss more in the appendix. Meanwhile, we make a few observations and open problems below.

**It may not be the end of the optimism [20].** Let us forget about CROP and consider the oracle described in Section 2. Consider $\mathcal{F}^{\text{code}}$ with $\frac{1}{2\epsilon} > \frac{2}{\Lambda^2}$. Note that UCB in fact has a regret bound of $O(\min\{\frac{K}{\epsilon} \ln(n), \epsilon n\})$; the first argument can be vacuous (i.e., $\geq n$) in which case we know the regret so far is $\epsilon n$ since UCB by design only pulls arm $i$ with $\Delta_i(f) = \epsilon$. The oracle has regret $\Theta(\min\{\frac{\ln(K)}{\Lambda^2} \ln(n), n\})$ where we have $n$ rather than $\epsilon n$ because she pulls informative arms. However, this implies that, until $n \lesssim \frac{1}{\Lambda^2}$, the oracle has a linear regret. In fact, all known algorithms with AO would be the same, to our knowledge. This is not just a theoretical observation. In Hao et al. [16, Figure 1], their algorithm with AO performs worse than an optimistic one until $n \approx 2000$. We ask if one can achieve the minimum of the two; i.e., obtain a finite-time regret bound of $O(\min\{\frac{\ln(K)}{\Lambda^2} \ln(n), \epsilon n\})$. This is a reminiscent of the "sub-UCB" criterion by Lattimore [18] (also discussed in Tirinzoni et al. [27]) in the sense that we like to perform no worse than UCB. For $\mathcal{F} = \mathcal{F}^{\text{code}}$, we provide a positive answer in the appendix, but a more generic algorithm that enjoys the AO and performs no worse than UCB for any $\mathcal{F}$ is an open problem.

**The worst-case regret.** For more on the worst-case regret and how it is different from the instance-dependent regret, see our related work section in the appendix. The example above shows that the oracle suffers a linear worst-case regret over the family of problems $\{\mathcal{F}^{\text{code},\epsilon,\Lambda} : \epsilon, \Lambda \leq (0, 1/2]\}$. That is, for any given problem complexity $n$ and $K$, one can always find $\epsilon$ and $\Lambda$ for which the oracle suffers a linear regret. This is in stark contrast to UCB that has $\widetilde{O}\left(\sqrt{Kn}\right)$ worst-case regret over this family. In fact, the oracle suffers a linear regret in linear bandits, too. In Example 4 of Lattimore and Szepesvári [20], it is easy to see that the oracle has regret $\min\{2\alpha^2 \ln(n), n\}$ when $\epsilon$ satisfies $2/\epsilon > 2\alpha^2$. Thus, given $n$, this bandit problem with $\alpha \approx \sqrt{n}$ for some $\epsilon$ with $2/\epsilon > 2\alpha^2$ would make the oracle suffer a linear regret for the instance $\theta = (1, 0)$. To our knowledge, all known AO algorithms share the same trait as they do not have any device to purposely avoid pulling the informative arm in the small-$n$ regime.

We believe the issue is not that we study instance-dependent regret but that we tend to focus too much on the leading term w.r.t. $n$ in the asymptotic regime, which we attribute to the fact that it is the one where the optimality can be claimed as of now. Less is known about the optimality on the lower order terms along with other instance-dependent parameters. This is studied a bit more in pure exploration problems [26, 17]. We hope to see more research on precise instance-dependent regret bounds in nonasymptotic regimes and practical structured bandit algorithms.

## Broader Impact

Our study is mainly about a novel approach to solve structured bandits algorithms where we try to overcome some shortcomings of existing methods. Algorithmic developments in bandits have a huge impact in many potential applications including dose-finding trials. In this application, a structured bandits that encode a proper inductive bias and can help resolve health issues of many people by significantly reducing time/trials needed to find dosage or the right types of drugs, leading to maximum efficacy with minimal side-effects.

## Acknowledgments

We thank the anonymous reviewers, Lalit Jain, Kevin Jamieson, Akshay Krishnamurthy, Tor Lattimore, Robert Nowak, Ardhendu Tripathy, and the organizers and participants of RL Theory Virtual Seminars for providing valuable feedback and helpful discussions. This work is supported in part by the startup fund from the University of Arizona.

## Footnotes

[1]Some algorithms like OSSB [10] parameterize the exploration rate as $\epsilon$, introducing $\epsilon K g_n$ for some $g_n = \omega(1)$ in the regret bound. One may attempt to set $\epsilon = 1/K$ to remove the dependence, but there is another term $K/\epsilon$ in the bound (see [10, Appendix 2.3]). Above all, we believe the dependence on $K$ has to appear somewhere in the regret if forced sampling is used.

[2]Concurrent studies by Degenne et al. [12] and Saber et al. [24] avoid forced sampling but still have an explicit linear dependence on $K$ in the regret.

[3]Our algorithms and theorems can be easily extended to the setting where optimal actions w.r.t. $f$ can be non-unique. This requires us to redefine the equivalence relationship, which we omit for brevity.

[4]OSSB in fact does not find the ERM but rather uses the empirical means of the arms to solve the optimization problem (2), which can work for some problem families. Still, we believe extending OSSB to use the ERM with suitable loss function should achieve (near) asymptotic optimality for the finite $\mathcal{F}$.

[5]Similar aggressive definitions of confidence sets have also appeared in recent works for other purposes [e.g. 20].

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
