[Supplementary Material]

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

# Appendix

## Table of Contents

## A  Related work

Structured bandits, generally defined, consist of bandit problems where there exist pieces of side-information or constraints in (and among) the mean rewards. While, of course, the standard $K$-armed bandit problem [38] is a special case, researchers usually use the term structured bandits for nontrivial structure (e.g., beyond simple constraints like having mean rewards in $[0, 1]$). We group related work by the kinds of guarantees each study aim to achieve.

**Worst-case regret bounds.**  Suppose we are given a predefined family of problems $\Gamma = \{\mathcal{F}\}$, The worst-case regret bound of an algorithm $\pi$ over the family $\Gamma$ is the one that answers the following question: given a set of problem complexity parameters like $n$ and $K$ (and $d$ in linear bandits, for example), what is the largest regret bound that $\pi$ can suffer over the family $\Gamma$? As an example, for the linear bandit problem with a fixed arm set, the family $\Gamma$ contains any bandit problem $\mathcal{F}$ with an arm set $\mathcal{A}$ and a feature map $v : \mathcal{A} \to \mathbb{R}^d$ for which every $f \in \mathcal{F}$ and $a \in \mathcal{A}$ satisfy $f(a) = \langle \theta, v(a) \rangle, \forall a \in \mathcal{A}$ for some $\theta \in \mathbb{R}^d$.

Many works in the structured bandits focus on specific parametric reward models, such as linear and generalized linear models [35, 1, 14, 17], and regret bounds of order $\sqrt{n}$ has been obtained in these settings. Russo and Van Roy [36] propose a general notion called eluder dimension that facilitates analysis for structured bandits for general function classes. For nonlinear reward models, worst-case regret guarantees have been obtained on reward structures such as unimodality [43], convexity [2], and Lipschitzness [25].

**Instance-dependent regret bounds.**  Instance-dependent regret bounds aim to capture finer structures of problem instances beyond the complexity of hypothesis classes. In the asymptotic regime (i.e., fixing a problem instance and letting time horizon $n$ go to infinity), many works [3, 20, 29, 22] derive matching asymptotic regret upper and lower bounds for *uniformly-good* algorithms (defined in Section 2). However, their analysis cannot be easily converted to obtain a finite-sample guarantee.

In the finite-sample regime, instance-optimal algorithms under specific model classes have been developed, such as unimodal reward [12] and Lipschitz rewards [31]. Under general reward function classes, Combes et al. [13] provide an asymptotically optimal algorithm that is amenable to non-asymptotic analyses, which is later extended to reinforcement learning [33]. However, their finite

sample guarantees depend strongly on the size of the action space due to the forced sampling as discussed in Section 4, although such a dependence goes away in the asymptotic regime.

We remark that the forced sampling is not bad when the problem is *unstructured* because the sampling complexity must scale with $K$ anyways. For example, in the pure-exploration version of the bandit problem [7, 8], the algorithm by Garivier and Kaufmann [19] relies on forced sampling, but there is no known evidence that the forced sampling degrades its performance while without it their algorithm can provably fail.

**Other instance-dependent regret bounds.** Many algorithms do not achieve the asymptotic optimality (not even within a constant factor) but possess instance-dependent guarantees in structured bandits, which is mostly based on the optimism, defined in (3), or often elimination (e.g., see Jamieson [23, Section 6] for linear bandits and Tirinzoni et al. [39] for generic structure). We remark that, unlike the standard $K$-armed bandits without structure, elimination-based approaches in structured bandits can have a different regret bound from that of the optimism (i.e., one is better than the other and vice versa depending on the problem instance) as described in Tirinzoni et al. [39].

Many linear bandit studies obtain regret guarantees that depend on the gap $\Delta$ between the best mean reward and the second-best mean reward [14, 17, 1]. As another example, in Lipschitz bandits, regret guarantees that depend on the zooming dimension or near-optimality dimension have been shown [24, 9]. Other particular structures including univariate linearity [32], global bandits [4, 37], regional bandits [42], and $K$-armed bandits with side-information [10, 40, 15] show that it is possible to achieve bounded regret.

For the generic structures, many studies provide finite-time instance-dependent guarantees that do not achieve a constant-factor asymptotically optimality [28, 6, 39, 21], except for known special cases where the reward function class has a factorized representation across different arms, i.e., in the settings of Lai and Robbins [26], and Bernetas and Katehakis [11]. These studies, however, have regret bounds that reflect the structure of the instance beyond the gap $\Delta$ mentioned above. We conjecture that the suboptimality in their bounds is rooted in their confidence sets that are an intersection of confidence intervals for each arm, let alone their sampling strategies. In contrast, CROP maintains a confidence set that captures the structure of the hypothesis class, which, as we show in the proof, allows us to connect the constraints of $c(f)$ in (2) to the concentration inequality and thus to the confidence set as well.

Although we focus on the finite hypothesis space for simplicity, our ultimate goal of the paper is to find a generic algorithmic principle for any hypothesis space. Our algorithm CROP achieves the asymptotic optimal regret with a constant-factor, enjoys bounded regret whenever possible, and has mild dependence on $K$, all thanks to our novel forced-sampling-free design. This partially resolves the open question raised by Tirinzoni et al. [39, Section 7] where they ask if one can design confidence-based strategies (as opposed to solving the optimization problem (2) along with forced sampling) that are optimal for general structures with good finite-time performance. We believe that such an advancement is not an artifact of the finite hypothesis space setting but the fact that we rely on the pessimism, the key novelty in our algorithm design.

## B  Details of the arguments

In this section, we provide more explanations on our argument in the main body in the order they appear, section by section.

### B.1  More on problem definition and preliminaries

Let us elaborate more on the cheating code example in Section 2.

**The cheating code example.** Let us elaborate more on the cheating code example in Section 2. To understand the optimization problem $c(f)$ for the cheating code class, let us consider a simple case where $K_0 = 2$ and $k = 1$. Without loss of generality, let $f^* = (1, 1 - \epsilon, 0)$. In this case, the only

competing hypothesis is $(1, 1 + \epsilon, \Lambda)$. Then, $c(f^*)$ is written as follows:

$$c(f^*) = \min_{\gamma_1 = 0, \gamma_2 \geq 0, \gamma_3 \geq 0} \quad \epsilon \gamma_2 + 1 \cdot \gamma_3$$
$$\text{s.t.} \quad \gamma_2 \frac{(2\epsilon)^2}{2} + \gamma_3 \frac{\Lambda^2}{2} \geq 1$$

One can see that a solution $\gamma_2 = \alpha \cdot \frac{1}{(2\epsilon)^2/2}$ and $\gamma_3 = (1 - \alpha) \cdot \frac{1}{\Lambda^2/2}$ for some $\alpha \in [0, 1]$ is feasible and satisfies the constraint with equality. Furthermore, one can see that any feasible solution that cannot be expressed by the solution above has a strictly larger objective value due to $\gamma \geq 0$ and $\epsilon > 0$. The objective function is then $\alpha \cdot \frac{1}{2\epsilon} + (1 - \alpha) \cdot \frac{2}{\Lambda^2}$. This means that whenever $\frac{1}{2\epsilon} > \frac{2}{\Lambda^2}$, setting $\alpha = 0$ achieves the minimum.

For generic $K_0$, we first provide an example of $f^* = (1, 1 - \epsilon, 1 - \epsilon, 1 - \epsilon, 0, 0)$ for the case of $K_0 = 4$.

$$c(f^*) = \min_{\gamma_1 = 0, \gamma_{2:K} \in (0, \infty]^{K-1}} \quad \epsilon \cdot \left( \sum_{a=2}^{K_0} \gamma_a \right) + 1 \cdot \left( \sum_{a=K_0+1}^{K} \gamma_a \right)$$
$$\text{s.t.} \quad \gamma_2 \frac{(2\epsilon)^2}{2} \qquad\qquad\qquad\qquad +\gamma_6 \frac{\Lambda^2}{2} \geq 1 \qquad (7)$$
$$+\gamma_3 \frac{(2\epsilon)^2}{2} \qquad\qquad +\gamma_5 \frac{\Lambda^2}{2} \qquad\qquad \geq 1$$
$$+\gamma_4 \frac{(2\epsilon)^2}{2} \quad +\gamma_5 \frac{\Lambda^2}{2} \quad +\gamma_6 \frac{\Lambda^2}{2} \geq 1$$

We use shorthands $\gamma_{i:j}$ for $\gamma_i, \gamma_{i+1}, \ldots, \gamma_j$. It is now nontrivial to see how the optimal solution would look like. The following proposition provides a characterization of the optimal solution.

**Proposition 1.** *Consider $\mathcal{F}^{code}$. Let $K_0 = 2^k$ for some integer $k$. We claim that if $\frac{1}{2\epsilon} > \frac{2}{\Lambda^2}$, then the solution of the optimization problem* (2) *is*

$$\gamma_a^\dagger = \begin{cases} 0, & a \in [K_0] \\ \frac{2}{\Lambda^2}, & a \in \{K_0 + 1, \ldots, K\} \end{cases}. \qquad (8)$$

*Proof.* For clarity, our convention is that the coordinate $K$ is for the least significant bit of the code. For example, when $K_0 = 4$, then the hypothesis $f$ with $a^*(f) = 2$ is $f = (1, 1 + \epsilon, 1 - \epsilon, 1 - \epsilon, 0, \Lambda)$.

The plan is to suppose that $u$ is a feasible solution to the optimization problem $c(f^*)$ for $\mathcal{F}^{code}$ for which there exists a coordinate $a \in [K_0] \setminus \{a^*\}$ with $u_a > 0$. Then, we show:

- First, we prove that it is possible to construct a feasible solution $v$ that is strictly better than $u$ where $v$ does not have nonzero entries for the first $K_0$ coordinates; this proves that the optimal solution must be supported only on the cheating arms.
- Second, we show that $\gamma^\dagger$ is the optimal solution.

As a starter, consider the example of (7). Suppose we have a feasible solution $u = (0, \frac{1}{4\epsilon^2}, 0, 0, \frac{2}{\Lambda^2}, \frac{1}{\Lambda^2})$. Then, the coordinate $q = 2$ is nonzero. Consider a hypothesis $g$ that has arm $q$ as the best arm: $g = (1, 1 + \epsilon, 1 - \epsilon, 1 - \epsilon, 0, \Lambda)$. The arm 6 is the cheating arm whose mean reward is different from that of $f^*$; let $j = 2$ so that $6 = K_0 + j$. Then, we can modify $u$ by zeroing out $\gamma_q$ and adding more mass to $\gamma_{K_0+j}$ so that the first constraint of (7) is satisfied. This modification leads to $h^* = (0, 0, 0, 0, \frac{2}{\Lambda^2}, \frac{2}{\Lambda^2})$. Note that this operation does not make other constraints violated because the variable $\gamma_q$ appears in one constraint only, and adding more mass to $\gamma_{K_0+j}$ never harms. We now generalize this example.

Let $q \in [K_0] \setminus \{a^*\}$ satisfy $u_q > 0$. Consider $g \in \mathcal{C}^*$ with $a^*(g) = q$. Let $j \in [k]$ be the largest index $j$ for which $g(K_0 + j) \neq f^*(K_0 + j)$ whose existence is certified by the definition of $\mathcal{F}^{code}$. Let $e_i$ be the $i$-th indicator vector. Let $\delta = \frac{2\epsilon^2 u_q}{\Lambda^2/2}$. We then define

$$h^*(u, q) = u - u_q e_q + \delta \cdot e_{K_0 + j}$$

One can show that our choice of $\delta$ indeed ensures that $h^*(u, q)$ is a feasible solution. We now show that this modified version $h^* := h^*(u, q)$ has a strictly smaller objective function value:

$$\left( \sum_{a=1}^{K_0} \epsilon u_a + \sum_{a=K_0+1}^{K} \Delta_a u_a \right) - \left( \sum_{a=1}^{K_0} \epsilon h_a^* + \sum_{a=K_0+1}^{K} \Delta_a h_a^* \right) = \epsilon u_q - \Delta_j \delta$$

$$\geq \epsilon u_q - \delta \qquad\qquad (\because \Delta_j \leq 1)$$

$$= \epsilon u_q - \frac{2\epsilon^2 u_q}{\Lambda^2/2}$$

$$= \epsilon u_q \left(1 - \frac{2\epsilon}{\Lambda^2/2}\right)$$

$$> 0 \quad (\because \text{ the assumption of the proposition})$$

Therefore, one can perform the following one sweep of coordinate descent.

---

- Input: $f^*$, $u$: a feasible solution of $c(f^*)$.
- $v^{(1)} \leftarrow u$
- For $a = 1, \ldots, K_0$,
    - $v^{(a+1)} \leftarrow h^*(v^{(a)}, a)$
- Output: $v^* := v^{(K_0+1)}$

---

We conclude the first part of the proof by the following observations:

- The output $v^*$ above has a strictly smaller objective function than that of the input $u$ whenever $\exists a \in [K_0] : u_a > 0$.
- The output $v^*$ satisfies that $v^*_{1:K_0} = 0$.

We now show the second part of the proof. Let $\gamma$ be a feasible solution that is supported only on the cheating arms. We claim that $\forall j \in [k]$, the value $\gamma_{K_0+j}$ must be at least $\frac{2}{\Lambda^2}$.

The reason is that, given $j \in [k]$, we can find $g \in \mathcal{C}^*$ for which $(g(K_0 + 1), \ldots, g(K))$ differs from $(f^*(K_0 + 1), \ldots, f^*(K))$ at the coordinate $j$ only, by the definition of $\mathcal{F}^{\text{code}}$; i.e., the binary representation of $a^*(g) - 1$ differs from that of $a^* - 1$ only at the $(k - j + 1)$-th least significant bit. In the example of (7), for $j = 1$, $g = (1, 1+\epsilon, 1-\epsilon, 1-\epsilon, 0, \Lambda)$ and for $j = 2$, $g = (1, 1-\epsilon, 1+\epsilon, 1-\epsilon, \Lambda, 0)$. Then, the constraint induced by that hypothesis $g$ is:

$$\gamma_{a^*(g)} \frac{(2\epsilon)^2}{2} + \gamma_{K_0+j} \frac{\Lambda^2}{2} \geq 1.$$

Plugging the fact that $\gamma_{a^*(g)} = 0$, we have $\gamma_{K_0+j} \frac{\Lambda^2}{2} \geq 1$, implying that $\gamma_{K_0+j} \geq \frac{2}{\Lambda^2}$. This proves the claim above, establishing a coordinate-wise lower bound on $\gamma$.

We observe that the objective function of this lower bound on $\gamma$ is a lower bound for the optimal solution, which is achieved by $\gamma^\dagger$; this concludes the proof. $\qquad\square$

## B.2 More on CROP

We here discuss more on the design choices for CROP. We do not claim our design of $\psi$ and $\phi$ is the best; fine-tuning of those and the algorithm itself is left as future work. In what follows, we focus on describing our intention on the current design *at the time of development*.

**The design of $\psi(f)$ in Fallback.** The reason why we now allow $\gamma_{a^*(f)} \neq 0$ is for the following case where $\epsilon > 0$ and we boldface the best arms:

| Arms  | A1   | A2              | A3              |
|-------|------|-----------------|-----------------|
| $f_1$ | **1**| .25             | .25             |
| $f_2$ | **.75**| .25           | .25             |
| $f_3$ | 0    | $.25 - 2\epsilon$ | **$.25 - \epsilon$** |
| $f_4$ | 0    | **.25**         | 0               |

Suppose $f^* = f_2$ and $\epsilon > 0$ is small enough. At the beginning, $\mathcal{F}_t = \{f_1, f_2, f_3, f_4\}$, $\widetilde{\mathcal{F}}_t = \{f_1\}$, $\overline{f}_t = \{f_3\}$. If we do not allow $\gamma_{a^*(f)}$ to be nonzero in (4), then $\psi(\overline{f}_t)$ will assign a very large number of pulls to arm A2 whereas pulling A3 will eliminate $f_4$ quickly. On the other hand, those troublesome hypotheses $f_3$ and $f_4$ belong to the docile class w.r.t. $f_2$, and incurring high regret for those (albeit finite terms) seems unreasonable.

Figure 3: A cartoon showing the regret bound of UCB and AO where AO is the asymptotically optimal oracle described in Section 2. The minimum of the two curves is shaded. Can we achieve this regret bound?

We have a constraint $\gamma \geq \phi(f) \vee \gamma(f)$ in (4), which we call the *extension* constraint. The intention is, if $f$ is the ground truth, we will have to make $\gamma_a(f) \ln(n)$ pulls for the informative arms $\{a : \gamma_a(f) > 0\}$ anyways, so we add it to the constraint. This guards against to the case where we inadvertently pull noninformative arms $\{a : \gamma_a(f) = 0\}$ too much. Our understanding is that, in general, this does not affect the regret bound too much, at least in the current analysis.

### B.3 More on analysis

The proof of Theorem 1 can be found in Appendix C.

**Remark 1.** Note we have a $\ln(|\mathcal{F}|)$ dependence, which comes from the naive union bounds. One can extends CROP to a larger or even infinite hypothesis space using the covering number argument as done in Foster et al. [18, Lemma 4], although we have focused on the finite hypothesis for simplicity.

### B.4 More on discussion

We provide missing details from the discussion section.

**Regret no more than the optimism.** Consider $\mathcal{F}^{\text{code}}$. Assume $f^* = (1, 1 - \epsilon, \ldots, 1 - \epsilon, 0, \ldots, 0) \in \mathcal{F}^{\text{code}}$ without loss of generality, and assume $\frac{1}{2\epsilon} > \frac{2}{\Lambda^2}$. The goal is to achieve a regret bound of $\mathbb{E} \operatorname{Reg}_n = O\left(\min\left\{\frac{\ln(K)}{\Lambda^2} \ln(n), \epsilon n\right\}\right)$, which is depicted in Figure 3. Note that, in the fixed budget setting (i.e., the target $n$ is given before running the algorithm), it is trivial to perform no worse than the optimism. Specifically, check the regret bound of the oracle or those that mimic the oracle (simply call them the oracle, hereafter) and see if it is larger than $\epsilon n$. If true, then simply pull any of the first $K_0$ arms uniformly throughout; otherwise, invoke the oracle.

For achieving the goal in the anytime setting, note that the optimism, such as UCB1 [5] run without knowing the structure or UCB-S [27], achieves an anytime regret bound of

$$\mathbb{E} \operatorname{Reg}_n \leq \min\left\{c_1 \frac{K}{\epsilon} \ln(n), \ \epsilon n\right\}$$

for $n \geq 2$ and some numerical constant $c_1$. The oracle achieves an anytime regret bound of

$$\mathbb{E} \operatorname{Reg}_n \leq \min\left\{c_2 \frac{\ln(K)}{\Lambda^2} \ln(n), \ n\right\}$$

for $n \geq 2$ and some numerical constant $c_2$.

Suppose $K$ is large enough so that $c_1 \frac{K}{\epsilon} > c_2 \frac{\ln(K)}{\Lambda^2}$. Let $t_0$ be the $t$ such that $c_2 \frac{\ln(K)}{\Lambda^2} \ln(t) = \epsilon t$, the time step after which the oracle outperforms UCB. The idea is simple: run UCB up to time step $t \leq t_0$ and then start the oracle as if we are starting from the beginning (in words, throw away all the samples so far). Then,

$$\mathbb{E} \operatorname{Reg}_n = \mathbb{1}\{n \leq t_0\}\epsilon n + \mathbb{1}\{n > t_0\} \mathbb{E}\left[\epsilon t_0 + c_2 \frac{\ln(K)}{\Lambda^2} \ln(n - t_0)\right]$$

$$\leq \mathbb{1}\{n \leq t_0\}2 \cdot \epsilon n + \mathbb{1}\{n > t_0\}2 \cdot c_2 \frac{\ln(K)}{\Lambda^2} \ln(n)$$

$$\le 2 \cdot \min \left\{ c_2 \frac{\ln(K)}{\Lambda^2} \ln(n), \epsilon n \right\} ,$$

which achieves our goal.

Note, however, that this simple strategy was possible because we knew $\epsilon$ for the given class $\mathcal{F}^{\text{code}}$. Consider $\mathcal{F}^{\text{code2}} := [1/2, 1]^{K_0} \times \{0, \Lambda\}^k$ where the last $k = \lceil \log_2(K_0) \rceil$ arms are binary codes encoding the best arm index (break ties with the smallest index). Then, the class contains hypotheses that have $\epsilon$ gap for the first $K_0$ arms for any $\epsilon$. Even if we do know that $f^*$ is one of those cases, we do not know $\epsilon$ and thus have to adaptively decide when to start pulling the informative arms via the rewards collected throughout. Achieving the asymptotic optimality while maintaining the standard worst-case regret (e.g., $\sqrt{dn\ln(K)}$ for linear bandits) seems to be an interesting open problem.

**Future work.** Our study unlocks numerous open problems besides achieving both the worst-case regret and asymptotic optimality. Since many bandit studies have focused on the leading term $\ln(n)$, the optimal sampling strategy that minimize the arm pulls of those noninformative arms is not studied well. We like to study the optimality of the number of arm pulls of noninformative arms w.r.t. not just $n$ but also problem-dependence parameters such as information gaps, which tend to matter when the number of arms is very large. Towards practical algorithms, we believe this is more important than getting the exact asymptotic optimality. Furthermore, we like to investigate if we can extend our pessimism to more popular structures such as linear bandits or Lipschitz bandits so one can achieve the asymptotic optimality without forced sampling (and without the dependence on the number of arms).

## C Upper bound proof

In this section, we discuss the proof of Theorems 1 and auxiliary lemmas for it.

**Theorem 3.** *Let $(\alpha, \mathring{\alpha}, z, \mathring{z}) = (2, 3, |\mathcal{F}|, |\mathcal{F}|)$. Suppose we run CROP with hypothesis class $\mathcal{F}$ with the environment $f^* \in \mathcal{F}$. Then, CROP has the following anytime regret guarantee: $\forall n \ge 2$,*

$$\mathbb{E}\,\mathrm{Reg}_n \le P_1 \ln n + c_1 \left( P_1 \left( \sqrt{\ln n \ln |\mathcal{F}|} + \ln\ln n + \ln|\mathcal{F}| \right) + P_2 \ln(\ln(n)) + P_3 \left( \ln(|\mathcal{F}|) + \ln(Q_1) \right) + K_\psi \Delta_{\max} \right),$$

*where $c_1$ is a numerical constant, and*

$$P_1 = \sum_a \Delta_a \gamma_a^*, \quad P_2 = \sum_a \Delta_a \phi_a(\mathcal{E}^*), \quad P_3 = \sum_a \Delta_a \psi_a(\mathcal{F})$$

*and $Q_1 = \Lambda_{\min}^{-2} + K_\psi(1 + \max_a \psi_a(\mathcal{F}))$. Furthermore, when $\gamma^* = 0$, we have $P_1 = P_2 = 0$, achieving a bounded regret.*

Before going into details we highlight some of the technical aspects of our proof. In our proof, Lemma 8 plays a key role for analyzing the confidence set $\mathring{\mathcal{F}}_t$ that has an aggressive confidence level, which we believe is not commonly dealt with in the standard bandit settings. One can see why this is needed in Section C.5.2 and Section C.5.3 where we apply the "regret peeling." The proofs in Section C.5.4 are a bit lengthy, but the main idea stems from Lattimore and Munos [27]. Other proofs are relatively standard, we believe.

We first present some martingale concentration inequalities for our confidence set.

### C.1 Concentration inequalities

This section establishes a few important concentration inequalities on the losses of reward regressors, which are instrumental in our analysis.

**Additional notations.** Recall that at time step $t$, the learner pulls arm $a_t$ and receives reward $r_t = f_*(a_t) + \xi_t$, where $\xi_t$ is $\sigma^2$-sub-Gaussian. To avoid double subscripting, we will sometimes use $a(t)$ to denote $a_t$. Define the instantaneous loss of regressor $f$ as time $s$ as $\ell_s(f) := (f(a_s) - r_s)^2$. At the end of time step $t$, we define the cumulative loss of $f$ as $L_t(f) := \sum_{s=1}^t \ell_s(f)$. Define the instantaneous regret of $f$ as $M_s(f) := \ell_s(f) - \ell_s(f^*)$. We define the *information* gap between $f$ and $f^*$ on action $a$ as: $\Lambda_a(f) := \frac{f(a(s)) - f^*(a(s))}{\sigma}$. A larger $\Lambda_a(f)$ implies that $f$ is easier to

be distinguished from $f^*$ by pulling arm $a$. This should not to be confused with the *reward* gap $\Delta_a(f) = \mu^*(f) - f_a$.

We define the filtration $\{\Sigma_t\}_{t=0}^{\infty}$ as follows: $\Sigma_t = \sigma(a_1, r_1, \ldots, a_t, r_t, a_{t+1})$. We abbreviate $\mathbb{E}_t[\cdot] = \mathbb{E}[\cdot \mid \Sigma_t]$. Define $\text{IC}(f, g, \pi) = \sum_{a \in \mathcal{A}} \pi(a) \frac{(f(a) - g(a))^2}{2\sigma^2}$ as the information constraint between $f$ and $g$ w.r.t. $\pi$. Define $\text{IC}^*(f, \pi) = \text{IC}(f, f^*, \pi)$.

Define $T(t) = (T_a(t))_{a \in \mathcal{A}}$ as the vector that encodes the number of arm pulls for each arm up to time step $t$. For vectors $u = (u_a)_{a \in \mathcal{A}}$, $v = (v_a)_{a \in \mathcal{A}}$, we denote $u \ge v$ if $\forall a \in \mathcal{A}, u_a \ge v_a$. We use shorthands $a^* = a^*(f^*)$ and $\mu^* = \mu^*(f^*)$.

We establish fundamental concentration result on partial sums of $\{M_s(f)\}$.

**Lemma 4.** (i) For any $f$ and any $\beta > 0$,

$$\mathbb{P}\left(\exists t \in \mathbb{N} \,.\, L_t(f^*) - L_t(f) \ge \beta\right) \le \exp\left(-\frac{\beta}{2\sigma^2}\right).$$

(ii) For any $f$ and any $r > 0, \beta > 0$,

$$\mathbb{P}\left(\exists t \in \mathbb{N} \,.\, \text{IC}(f, f^*, T(t)) \ge r \,\wedge\, L_t(f) - L_t(f^*) \le \beta\right) \le \exp\left(-\frac{1}{4r}\left(r - \frac{\beta}{2\sigma^2}\right)_+^2\right) \quad (9)$$

$$\le \exp\left(-\frac{\sigma^2 r - \beta}{4\sigma^2}\right). \quad (10)$$

*Proof.* Throughout the proof, we will abbreviate $M_s(f)$ as $M_s$ to avoid notation clutter. It can be easily seen that $\sum_{s=1}^{t} M_s = L_t(f) - L_t(f^*)$. Define $d_s = \sigma\Lambda_{a(s)}(f) = f(a) - f^*(a)$.

The first item follows from Equation (12) in Lemma 5 with $\lambda = \frac{1}{\sigma^2}$ and $\delta = \exp\left(-\frac{\beta}{2\sigma^2}\right)$.

For the second item, we first show Equation (9). The equation is trivially true when $2\sigma^2 r < \beta$, so we focus on the case of $2\sigma^2 r \ge \beta$. With foresight, consider setting $\delta = \exp\left(-\frac{1}{4r}\left(r - \frac{\beta}{2\sigma^2}\right)^2\right)$ and $\lambda = \frac{1}{2\sigma^2}\sqrt{\frac{\ln\frac{1}{\delta}}{r}} = \frac{1}{4\sigma^2 r}\left(r - \frac{\beta}{2\sigma^2}\right)$ in Equation (12) in Lemma 5, we get

$$\mathbb{P}\left(\exists t \in \mathbb{N} \,.\, \sum_{s=1}^{t} M_s \le \left(1 - \sqrt{\frac{\ln\frac{1}{\delta}}{r}}\right)\left(\sum_{s=1}^{t} d_s^2\right) - 2\sigma^2\sqrt{r \ln\frac{1}{\delta}}\right) \le \delta.$$

As the setting of $\delta$ implies that $r \ge \ln\frac{1}{\delta}$, we have $\left(1 - \sqrt{\frac{\ln\frac{1}{\delta}}{r}}\right) \ge 0$. This implies that

$$\mathbb{P}\left(\exists t \in \mathbb{N} \,.\, \sum_{s=1}^{t} d_s^2 \ge 2\sigma^2 r, \sum_{s=1}^{t} M_s \le 2\sigma^2 r - 4\sigma^2\sqrt{r \ln\frac{1}{\delta}}\right) \le \delta.$$

The setting of $\delta$ implies that $2\sigma^2 r - 4\sigma^2\sqrt{r \ln\frac{1}{\delta}} = \beta$. In addition, observing that $\sum_{s=1}^{t} d_s^2 = 2\sigma^2 \text{IC}(f, f^*, T(t))$, and $\sum_{s=1}^{t} M_s = L_t(f) - L_t(f^*)$, the above inequality is precisely (9).

For Equation (10), it suffices to show $\frac{1}{4r}\left(r - \frac{\beta}{2\sigma^2}\right)_+^2 \ge \frac{\sigma^2 r - \beta}{4\sigma^2}$. This is trivially true if $\sigma^2 r < \beta$. Otherwise, $\sigma^2 r \ge \beta$; in this case, the above is a direct consequence of the AM-GM inequality that $\left(r - \frac{\beta}{2\sigma^2}\right)^2 \ge r \cdot \left(r - \frac{\beta}{\sigma^2}\right)$. $\qquad\square$

**Lemma 5.** *Suppose $f$ is in $\mathcal{F}$. Define $d_s(f) = \sigma\Lambda_{a(s)}(f) = f(a) - f^*(a)$. Then, for any $\lambda > 0$ and $\delta > 0$,*

$$\mathbb{P}\left(\exists t \in \mathbb{N} \,.\, \sum_{s=1}^{t} M_s(f) \ge (1 + 2\sigma^2\lambda)\sum_{s=1}^{t} d_s(f)^2 + \frac{1}{\lambda}\ln\frac{1}{\delta}\right) \le \delta, \quad (11)$$

$$\mathbb{P}\left(\exists t \in \mathbb{N}. \sum_{s=1}^{t} M_s(f) \le (1 - 2\sigma^2\lambda) \sum_{s=1}^{t} d_s(f)^2 - \frac{1}{\lambda} \ln \frac{1}{\delta}\right) \le \delta. \tag{12}$$

*Proof.* Throughout the proof, we will abbreviate $M_s(f)$ as $M_s$, and abbreviate $d_s(f)$ as $d_s$, to avoid notation clutter.

First, observe that $\ell_s(f^*) = \xi_s^2$. in addition,

$$M_s = \ell_s(f) - \ell_s(f^*) = (d_s - \xi_s)^2 - \xi_s^2 = d_s^2 - 2d_s\xi_s,$$

which implies that

$$\mathbb{E}_{s-1}[M_s] = d_s^2.$$

For any $\theta \in \mathbb{R}$, define $H_t = \exp\left(\theta \sum_{s=1}^{t} M_s - \theta(1 + 2\sigma^2\theta) \sum_{s=1}^{t} d_s^2\right)$ with the convention $H_0 = 1$. We now show $\{H_t\}_{t=0}^{n}$ is a nonnegative supermartingale.

By the calculations on $M_s$ above, $\theta M_s - \theta(1 + 2\sigma^2\theta)d_s^2 = -2\theta d_s\xi_s - 2\sigma^2\theta^2 d_s^2$. Using this, we have

$$\mathbb{E}_{t-1}[H_t] = \mathbb{E}_{t-1}\left[H_{t-1} \cdot \exp\left(\theta M_t - \theta(1 + 2\sigma^2\theta)d_t^2\right)\right] = H_{t-1} \cdot \mathbb{E}_{t-1}\left[\exp\left(-2\theta d_t\xi_t - 2\sigma^2\theta^2 d_t^2\right)\right].$$

Observe that by the $\sigma^2$-subgaussian property of $\xi_t$,

$$\mathbb{E}_{t-1}\left[\exp\left(-2\theta d_t\xi_t - 2\sigma^2\theta^2 d_t^2\right)\right] = \mathbb{E}_{t-1}\left[\exp\left(-2\theta d_t\xi_s\right)\right]\exp\left(-2\sigma^2\theta^2 d_t^2\right)$$

$$\le \exp\left(\frac{(2\theta)^2 d_t^2 \sigma^2}{2}\right)\exp\left(-2\sigma^2\theta^2 d_t^2\right) = 1.$$

This shows that $\mathbb{E}_{t-1}[H_t] \le H_{t-1}$, proving $\{H_t\}_{t=0}^{n}$ is a nonnegative supermartingale. Therefore, by Ville's maximal inequality [41] (see also Durrett [16, Exercise 5.7.1]), we have that for any $\delta > 0$,

$$\mathbb{P}(\exists t \in \mathbb{N}. H_t \ge \delta^{-1}) \le \delta.$$

Choosing $\theta = \lambda$, and noting that $H_t \ge \frac{1}{\delta}$ is equivalent to $\sum_{s=1}^{t} M_s \ge (1 + 2\sigma^2\lambda) \sum_{s=1}^{t} d_s^2 + \frac{1}{\lambda} \ln \frac{1}{\delta}$, we get Equation (11). Likewise, choose $\theta = -\lambda$, and noting that $H_t \ge \frac{1}{\delta}$ is equivalent to $\sum_{s=1}^{t} M_s \le (1 - 2\sigma^2\lambda) \sum_{s=1}^{t} d_s^2 - \frac{1}{\lambda} \ln \frac{1}{\delta}$, we get Equation (12). $\qquad\square$

**Lemma 6.** *Let* $\beta_t = 2\sigma^2 \ln(zt(\log_2 t)^2)$ *with* $z \ge |\mathcal{F}|$ *and* $\alpha \ge 1$. *Define* $B_t = \{L_{t-1}(f^*) - \min_{f \in \mathcal{F}} L_{t-1}(f) > \beta_t\}$. *Let* $q \ge 1$ *be an integer. Then,*

$$\mathbb{P}(\exists t \ge q : B_t) \le \frac{1}{q(\log_2 q)^2}.$$

*Proof.*

$$\mathbb{P}(\exists t \ge q : B_t) = \mathbb{P}(\exists t \ge q : L_{t-1}(f^*) - \min_{f \in \mathcal{F}} L_{t-1}(f) > \beta_t)$$

$$\le \mathbb{P}(\exists t \ge q : L_{t-1}(f^*) - \min_{f \in \mathcal{F}} L_{t-1}(f) > \beta_q)$$

$$\le \mathbb{P}(\exists t \in \mathbb{N}, f \in \mathcal{F} : L_{t-1}(f^*) - L_{t-1}(f) > \beta_q)$$

$$\le \sum_{f \in \mathcal{F}} \mathbb{P}(\exists t \in \mathbb{N} : L_{t-1}(f^*) - L_{t-1}(f) > \beta_q)$$

$$\le |\mathcal{F}| \cdot \exp\left(-\frac{\beta_q}{2\sigma^2}\right) \le \frac{1}{q(\log_2 q)^2}$$

where the first inequality is from the fact that $\beta_t$ is monotonically increasing; the second inequality is by relaxing the range of $t$; the third inequality is from union bound; the fourth inequality is from item (i) of Lemma 4; the last inequality is by algebra. $\qquad\square$

## C.2 Generic lemmas for analyzing bandit algorithms

The following is a standard inequality used in the UCB analysis.

**Lemma 7.** *Consider any bandit algorithm. Let $a_t$ be the index of the arm pulled at time $t$ and $T_i(t-1)$ be the number of times arm $i$ is pulled up to (and including) time $t-1$. Let $\tau$ be an integer and $Q_t$ be an event. Then,*

$$\sum_{t=1}^{n} \mathbb{1}\{a_t = i, Q_t\} \leq \tau + \sum_{t=\tau+1}^{n} \mathbb{1}\{a_t = i, Q_t, T_i(t-1) \geq \tau\}$$

*Proof.*

$$\sum_{t=1}^{n} \mathbb{1}\{a_t = i, Q_t\} = \sum_{t=1}^{n} \mathbb{1}\{a_t = i, Q_t, T_i(t-1) < \tau\} + \sum_{t=1}^{n} \mathbb{1}\{a_t = i, Q_t, T_i(t-1) \geq \tau\}$$

The first summation is bounded by $\tau$, for the following reason: if there are $\tau + 1$ time steps $t_1 < \ldots < t_{\tau+1}$ in which $a_t = i, Q_t, T_i(t-1) < \tau$ holds, we have that $T_i(t_{\tau+1} - 1) \geq \tau$, which contradicts with the fact that $T_i(t_{\tau+1} - 1) < \tau$.

Furthermore, for $t \leq \tau$, it must be the case that $T_i(t-1) \leq t-1 \leq \tau-1$, therefore the first $\tau$ terms of the second sum must be zero. This implies that the second sum equals $\sum_{t=\tau+1}^{n} \mathbb{1}\{a_t = i, Q_t, T_i(t-1) \geq \tau\}$. $\square$

In fact, the indicator terms in the lemma above have dependencies between different time steps that is not being explicitly captured. When we take the expectation and apply concentration inequalities, these dependencies are lost. The following lemma extends Lemma 7 so that such a dependency becomes explicit, which help prove tighter bounds. The basic idea is that whenever we pull an arm $a$ at time $t$, the count of arm $a$ increases by 1, so by the time we pull arm $a$, the pull count must be larger; this helps, when taking the expectation, obtain a tighter concentration of measure.

**Lemma 8.** *Under the same assumptions in Lemma 7,*

$$\sum_{t=1}^{n} \mathbb{1}\{a_t = i, Q_t\} \leq \tau + \sum_{m=1}^{\infty} \mathbb{1}\left\{\exists t \geq \tau + 1 : a_t = i, T_i(t-1) \geq \tau + m - 1, Q_t\right\}$$

*Proof.* By Lemma 7,

$$\sum_{t=1}^{n} \mathbb{1}\{a_t = i, Q_t\} \leq \tau + \sum_{t=\tau+1}^{n} \mathbb{1}\{a_t = i, Q_t, T_i(t-1) \geq \tau\}$$

Define $t^- = t - 1$ Define an event:

$$A_t = \left\{a_t = i, T_i(t^-) \geq \tau, Q_t\right\}$$

We aim to bound $\sum_{t=\tau+1}^{n} \mathbb{1}\{A_t\}$. Define $t_m$ to be the $m$-th time step after (and including) $t = \tau + 1$ that $A_t = 1$ is true; i.e.,

$$t_1 := \min\{t \in [\tau+1, n] : A_t \text{ is true }\}$$
$$\forall m \geq 2, t_m := \min\{t \in [t_{m-1}+1, n] : A_t \text{ is true }\}$$

where we take $\min \varnothing = \infty$. One can verify that, if $t_m < \infty$, then $T_i(t_m^-) \geq \tau + m - 1$. Then,

$$\sum_{t=\tau+1}^{n} \mathbb{1}\{A_t\} = \sum_{m=1}^{\infty} \mathbb{1}\{t_m < \infty\} \mathbb{1}\{A_{t_m}\}$$

$$= \sum_{m=1}^{\infty} \mathbb{1}\left\{t_m < \infty, a_{t_m} = i, T_i(t_m^-) \geq \tau, Q_{t_m}\right\}$$

$$\overset{(a)}{\leq} \sum_{m=1}^{\infty} \mathbb{1}\left\{t_m < \infty, a_{t_m} = i, T_i(t_m^-) \geq \tau + m - 1, Q_{t_m}\right\}$$

$$\leq \sum_{m=1}^{\infty} \mathbb{1}\left\{t_m < \infty, \exists t \geq \tau + 1 : a_t = i, T_i(t^-) \geq \tau + m - 1, Q_t\right\}$$

$$\leq \sum_{m=1}^{\infty} \mathbb{1}\left\{\exists t \geq \tau+1 : a_t = i, T_i(t^-) \geq \tau+m-1, Q_t\right\}$$

where $(a)$ is by our observation above. $\qquad\square$

## C.3  Lemmas on the execution of CROP

Depending on the execution trace of CROP, at time $t$, we define four events that form a disjoint union of the sample space:

1. 'E'xploit: $\mathrm{Ex}_t$, i.e. $a^*(\mathcal{F}_t)$ is singleton (line 4)
2. 'C'on'f'lict: $\mathrm{Cf}_t$ (line 12)
3. 'F'ea's'ibility: $\mathrm{Fs}_t$ (line 14)
4. 'F'all'b'ack: $\mathrm{Fb}_t$ (line 16)

The following lemma becomes useful when showing that, even if one utilizes $f$ to eliminate $g$ while neither $f$ nor $g$ is the ground truth, she will successfully eliminate either $f$ or $g$, up to a constant-factor w.r.t. the arm pulls.

**Lemma 9.** *For any $f, g$ and $\pi \geq 0$, we have $\mathrm{IC}(f, g, \pi) \leq \max(\mathrm{IC}^*(f, 4\pi), \mathrm{IC}^*(g, 4\pi))$.*

*Proof.* Given $\pi \geq 0$, it can be easily seen that $d(f,g) := \sqrt{\mathrm{IC}(f,g,\pi)} = \|\vec{f} - \vec{g}\|_{M_\pi}$ , where we denote $\vec{h} = (h_a)_{a\in\mathcal{A}}$ and $M_\pi = \mathrm{diag}(\frac{\pi_a}{2\sigma^2} : a \in \mathcal{A})$; therefore $d(f,g)$ is a Mahalanobis distance, hence satisfying triangle inequality. Specifically,

$$d(f,g) \leq d(f, f^*) + d(g, f^*) \leq 2\max(d(f, f^*), d(g, f^*)).$$

Squaring both sides, we have $d(f,g)^2 \leq 4\max(d(f,f^*)^2, d(g,f^*)^2)$, which implies that

$$\mathrm{IC}(f,g,\pi) \leq \max(\mathrm{IC}^*(f, 4\pi), \mathrm{IC}^*(g, 4\pi)). \qquad\square$$

**Lemma 10.** *In Algorithm 1, for every $t$, if we do not enter* Exploit*, then $\exists f \in \mathcal{F}_t : \mathrm{IC}^*(f, 4\pi_t) \geq 1$; furthermore, if $\mathrm{Cf}_t$ happens, then $\exists f \in \mathring{\mathcal{F}}_t : \mathrm{IC}^*(f, 4\pi_t) \geq 1$.*

*Proof.* There are three cases: $\mathrm{Cf}_t, \mathrm{Fs}_t, \mathrm{Fb}_t$.

1. If $\mathrm{Cf}_t$, then by the definition of $\phi(\overline{f}_t)$ and $\mathrm{Cf}_t$, there exists $f \in \mathring{\mathcal{F}}_t$ such that $\mathrm{IC}(f, \overline{f}_t, \phi(\overline{f}_t)) \geq 1$. As $\overline{f}_t$ is also in $\mathring{\mathcal{F}}_t$, by Lemma 9, $\exists f \in \mathring{\mathcal{F}}_t : \mathrm{IC}^*(f, 4\phi(\overline{f}_t)) \geq 1$. Therefore, we can state $\exists f \in \mathcal{F}_t : \mathrm{IC}^*(f, 4\phi(\overline{f}_t)) \geq 1$.
2. If $\mathrm{Fs}_t$, with a similar logic, either $\forall f \in \widetilde{\mathcal{F}}_t : \mathrm{IC}^*(f, 4\gamma(\overline{f}_t)) \geq 1$ or $\mathrm{IC}^*(\overline{f}_t, 4\gamma(\overline{f}_t)) \geq 1$. As $\widetilde{\mathcal{F}}_t \subseteq \mathcal{F}_t$ and $\overline{f}_t \in \mathcal{F}_t$, we have $\exists f \in \mathcal{F}_t : \mathrm{IC}^*(f, 4\gamma(\overline{f}_t)) \geq 1$ as well.
3. If $\mathrm{Fb}_t$, then with the same reasoning as the $\mathrm{Fs}_t$ case, either $\forall f \in \widetilde{\mathcal{F}}_t : \mathrm{IC}^*(f, 4\psi(\overline{f}_t)) \geq 1$ or $\mathrm{IC}^*(\overline{f}_t, 4\psi(\overline{f}_t)) \geq 1$, so we have $\exists f \in \mathcal{F}_t : \mathrm{IC}^*(f, 4\psi(\overline{f}_t)) \geq 1$ as well. $\qquad\square$

Recall that we use $a(t) := a_t$ to avoid double subscripts.

**Lemma 11.** *If Algorithm 1 enters state $\mathrm{Cf}_t$, $\mathrm{Fs}_t$, or $\mathrm{Fb}_t$, we have $\pi_t \neq 0$. In addition, $\pi_{a(t)} \neq 0$ holds with probability 1.*

*Proof.* We consider three cases:

- If $\mathrm{Cf}_t$, there must exist $g \in \mathcal{E}(\overline{f}_t) \setminus \{\overline{f}_t\}, \gamma(g) \not\subset \gamma(f)$. By the definition of $\phi$, we must have $\mathrm{IC}(\overline{f}_t, g, \phi(\overline{f}_t)) \geq 1$, implying that $\pi_t = \phi(\overline{f}_t) \neq 0$.
- If $\mathrm{Fs}_t$, then by the entering condition and the fact that $\widetilde{f}_t \in \widetilde{\mathcal{F}}_t$, $\mathrm{IC}(\overline{f}_t, \widetilde{f}_t, \gamma(\overline{f}_t)) \geq 1$ , implying that $\pi_t = \gamma(\overline{f}_t) \neq 0$.
- If $\mathrm{Fb}_t$, then by the definition of $\psi$ and the fact that $\mu^*(\widetilde{f}_t) \geq \mu^*(\overline{f}_t)$ and $a^*(\widetilde{f}_t) \neq a^*(\overline{f}_t)$, we must have $\forall \widetilde{f} \in \widetilde{\mathcal{F}}_t, \mathrm{IC}(\overline{f}_t, \widetilde{f}, \psi(\overline{f}_t)) \geq 1$, implying that $\pi_t = \psi(\overline{f}_t) \neq 0$.

For the other claim, if $\pi_{a(t)} = 0$, then we have $\frac{T_{a(t)}(t-1)}{\pi_{t,a(t)}} = \infty$, so the selection of $a(t)$ implies that $\pi_t = 0$, which is impossible by the first claim here. $\qquad\square$

We now present a useful lemma that formalizes the intuition that tracking (line 18) controls the number of arm pulls and thus the statistical power (i.e., information) to distinguish $f^*$ from the rest. We remark that the key variable below is $\zeta$, which appears three times in the LHS below.

**Lemma 12.** *Let $\zeta \in [0, \infty)^K$. Then, for any hypothesis $f$,*
$$\overline{\mathrm{Ex}_t}, \pi_t \propto \zeta, a_t = a, T_a(t-1) \geq \rho\zeta_a, \mathrm{IC}^*(f, \zeta) \geq c \implies \mathrm{IC}^*(f, T(t-1)) \geq \rho c .$$

*Proof.* By the definition of $a_t = \arg\min_a \frac{T_a(t-1)}{\pi_{t,a}} = \arg\min_a \frac{T_a(t-1)}{\zeta_a}$ and the condition that $T_a(t-1) \geq \rho\zeta_a$, we have, for every $b \in \mathcal{A}$,
$$\frac{T_b(t-1)}{\zeta_b} \geq \frac{T_a(t-1)}{\zeta_a} \geq \rho .$$
This implies that
$$T(t-1) \geq \rho\zeta. \tag{13}$$
As a consequence, $\mathrm{IC}^*(f, T(t-1)) \geq \mathrm{IC}^*(f, \rho\zeta) = \rho\mathrm{IC}^*(f, \zeta) \geq \rho c.$ $\qquad\square$

As an application of Lemma 12, we have the following lemma that will be useful for bounding the number of times different branches of CROP ($\mathrm{Cf}_t$, $\mathrm{Fs}_t$, and $\mathrm{Fb}_t$) are entered.

**Lemma 13.** *The following statements hold:*

(i) *For any $a \in \mathcal{A}$, $t \in \mathbb{N}$ and $\rho > 0$,*
$$\mathbb{P}\left(\exists s \in [t].\mathrm{Fs}_s, \pi_s \propto \gamma_a(f^*), a_s = a, T_a(s-1) \geq \rho\gamma_a(f^*), \widetilde{\mathcal{F}}_s \subseteq \mathcal{C}^*\right) \leq \exp\left(-\frac{1}{4\rho}\left(\rho - \frac{\beta_t}{2\sigma^2}\right)_+^2\right)$$

(ii) *For any $a \in \mathcal{A}$, $t \in \mathbb{N}$ and $\rho > 0$,*
$$\mathbb{P}\left(\exists s \in [t].\overline{\mathrm{Ex}_s}, a_s = a, T_a(s-1) \geq \rho\pi_{s,a}\right) \leq |\mathcal{F}| \cdot \exp\left(-\frac{\sigma^2\rho - 4\beta_t}{16\sigma^2}\right).$$

(iii) *For any $a \in \mathcal{A}$, $t \in \mathbb{N}$ and $\rho > 0$,*
$$\mathbb{P}\left(\exists s \in [t].\mathrm{Cf}_s, a_s = a, T_a(s-1) \geq \rho\pi_{s,a}\right) \leq |\mathcal{F}| \cdot \exp\left(-\frac{\sigma^2\rho - 4\mathring{\beta}_t}{16\sigma^2}\right).$$

*Proof.*

(i) If the event inside $\mathbb{P}(\cdot)$ happens, we have the following: there exists some $s_0 \in [t]$, such that $\mathrm{Fs}_{s_0}$ happens, $\pi_{s_0} \propto \gamma_a(f^*)$, $a_{s_0} = a$, $T_a(s_0 - 1) \geq \rho\gamma_a(f^*)$, $\widetilde{\mathcal{F}}_{s_0} \subseteq \mathcal{C}^*$. As $\widetilde{\mathcal{F}}_{s_0} \subseteq \mathcal{C}^*$, there must exists some $f_0 \in \mathcal{F}_{s_0}$ such that $\mathrm{IC}(f_0, f^*, \gamma(f^*)) \geq 1$. For this $f_0 \in \mathcal{F}_{s_0}$, $L_{s_0-1}(f_0) - \min_{g \in \mathcal{F}} L_{s_0-1}(g) \leq \beta_{s_0} \implies L_{s_0-1}(f_0) - L_{s_0-1}(f^*) \leq \beta_t$. Taking $\zeta = \gamma(f^*)$ in Lemma 12, we have $\mathrm{IC}(f_0, f^*, T(s_0 - 1)) \geq \rho$. Therefore,
$$\mathbb{P}\left(\exists s \in [t], \mathrm{Fs}_s, \pi_s \propto \gamma_a(f^*), a_s = a, T_a(s-1) \geq \rho\gamma_a(f^*), \widetilde{\mathcal{F}}_s \subset \mathcal{C}^*\right)$$
$$\leq \mathbb{P}\left(\exists s \in [t], f \in \mathcal{F}.L_{s-1}(f) - L_{s-1}(f^*) \leq \beta_t, \mathrm{IC}(f, f^*, T(s-1)) \geq \rho\right)$$
$$\leq \sum_{f \in \mathcal{F}} \mathbb{P}\left(\exists s \in [t].L_{s-1}(f) - L_{s-1}(f^*) \leq \beta_t, \mathrm{IC}(f, f^*, T(s-1)) \geq \rho\right)$$
$$\leq |\mathcal{F}| \exp\left(-\frac{1}{4\rho}\left(\rho - \frac{\beta_t}{2\sigma^2}\right)_+^2\right),$$

where the second inequality is from union bound, and the last inequality is from Equation (9) in Lemma 4(ii) and algebra.

(ii) If the event inside $\mathbb{P}(\cdot)$ happens, we have the following: there exists $s_0 \in [t]$, such that $\overline{\mathrm{Ex}}_{s_0}$ happens, $a_{s_0} = a$, and $T_a(s_0 - 1) \geq \rho\pi_{s_0,a}$. From Lemma 10, there exists $f_0 \in \mathcal{F}_{s_0} \subseteq \left\{ f : L_{s_0-1}(f) - L_{s_0-1}(f^*) \leq \beta_{s_0} \right\}$ such that $\mathrm{IC}(f_0, f^*, 4\pi_{s_0}) \geq 1$, implying that $\mathrm{IC}(f_0, f^*, \pi_{s_0}) \geq 1/4$. Taking $\zeta = \pi_{s_0}$ in Lemma 12, we have that $\mathrm{IC}(f_0, f^*, T(s_0 - 1)) \geq \rho/4$. Therefore,

$$
\begin{aligned}
&\mathbb{P}(\exists s \in [t]. \overline{\mathrm{Ex}}_s, a_s = a, T_a(s-1) \geq \rho\pi_{s,a}) \\
&\leq \mathbb{P}(\exists s \in [t], f \in \mathcal{F}. L_{s-1}(f) - L_{s-1}(f^*) \leq \beta_s, \mathrm{IC}(f, f^*, T(s-1)) \geq \rho/4) \\
&\leq \mathbb{P}(\exists s \in [t], f \in \mathcal{F}. L_{s-1}(f) - L_{s-1}(f^*) \leq \beta_t, \mathrm{IC}(f, f^*, T(s-1)) \geq \rho/4) \\
&\leq \sum_{f \in \mathcal{F}} \mathbb{P}(\exists s \in [t]. L_{s-1}(f) - L_{s-1}(f^*) \leq \beta_t, \mathrm{IC}(f, f^*, T(s-1)) \geq \rho/4) \\
&\leq |\mathcal{F}| \exp\left( -\frac{\sigma^2 \rho - 4\beta_t}{16\sigma^2} \right),
\end{aligned}
$$

where the second inequality uses the fact that $\beta_s \leq \beta_t$ for all $s \in [t]$; the third inequality is by union bound; the last inequality follows from Equation (10) of Lemma 4(ii) and algebra.

(iii) If the event inside $\mathbb{P}(\cdot)$ happens, we have the following: there exists $s_0 \in [t]$, such that $\mathrm{Cf}_{s_0}$ happens, $a_{s_0} = a$, and $T_a(s_0 - 1) \geq \rho\pi_{s_0,a}$. As $\mathrm{Cf}_{s_0}$ happens, by Lemma 10, there exists $f_0$ in $\mathring{\mathcal{F}}_{s_0} \subseteq \left\{ f \in \mathcal{F} : L_{s_0-1}(f) - L_{s_0-1}(\overline{f}_{s_0}) \leq \mathring{\beta}_{s_0} \right\}$ such that $\mathrm{IC}(f_0, f^*, 4\pi_{s_0}) \geq 1$. Taking $\zeta = \pi_{s_0} = \phi(\overline{f}_{s_0})$ in Lemma 12, we have for that $f_0$, $\mathrm{IC}(f_0, f^*, T(s_0-1)) \geq \rho/4$. Therefore,

$$
\begin{aligned}
&\mathbb{P}(\exists s \in [t]. \mathrm{Cf}_s, a_s = a, T_a(s-1) \geq \rho\pi_{s,a}) \\
&\leq \mathbb{P}(\exists s \in [t], f \in \mathcal{F}. L_{s-1}(f) - L_{s-1}(f^*) \leq \mathring{\beta}_s, \mathrm{IC}(f, f^*, T(s-1)) \geq \rho/4) \\
&\leq \mathbb{P}(\exists s \in [t], f \in \mathcal{F}. L_{s-1}(f) - L_{s-1}(f^*) \leq \mathring{\beta}_t, \mathrm{IC}(f, f^*, T(s-1)) \geq \rho/4) \\
&\leq \sum_{f \in \mathcal{F}} \mathbb{P}(\exists s \in [t]. L_{s-1}(f) - L_{s-1}(f^*) \leq \mathring{\beta}_t, \mathrm{IC}(f, f^*, T(s-1)) \geq \rho/4) \\
&\leq |\mathcal{F}| \exp\left( -\frac{\sigma^2 \rho - 4\mathring{\beta}_t}{16\sigma^2} \right).
\end{aligned}
$$

where the second inequality uses the fact that $\mathring{\beta}_s \leq \mathring{\beta}_t$ for all $s \in [t]$; the third inequality is by union bound; the last inequality follows from Equation (10) of Lemma 4(ii) and algebra. $\quad\square$

## C.4 Main proofs

Recall that $f^*$ is the ground truth mean rewards unless mentioned otherwise. Throughout, we use shorthands for the ground truth: $a^* := a^*(f^*)$, $\mu^* := \mu^*(f^*)$, and $\Delta_a := \Delta_a(f^*)$.

Recall that we have define $\psi(\mathcal{G})$, $\phi(\mathcal{G})$ and $K_\psi$ in the main text. Throughout we frequently use the notation $\mathcal{A}_\zeta = \{a \in \mathcal{A} : \zeta_a > 0\}$ for vector $\zeta \in [0, \infty)^K$.

Unlike observable states such as $\mathrm{Ex}_t$, $\mathrm{Cf}_t$, $\mathrm{Fs}_t$, and $\mathrm{Fb}_t$, there are hidden states that become useful for the purpose of analysis. Based on the relationship between the hypothesis sets constructed by CROP and the hypothesis classes related to the ground truth hypothesis $f^*$, we define four *hidden states* of CROP:

1. 'B'ad: : $B_t = \{ f^* \notin \mathcal{F}_t \}$.
2. Strongly steady state: $S_t^+ = \{ \widetilde{\mathcal{F}}_t \subseteq \mathcal{C}^*, \overline{f}_t \in \mathcal{E}^*, \gamma(\overline{f}_t) \propto \gamma^* \}$.
3. Weakly steady state: $S_t^0 = \{ \widetilde{\mathcal{F}}_t \subseteq \mathcal{C}^*, \overline{f}_t \in \mathcal{E}^*, \gamma(\overline{f}_t) \not\propto \gamma^* \}$.
4. Non-steady state: $S_t^- = \{ \widetilde{\mathcal{F}}_t \not\subseteq \mathcal{C}^* \vee \overline{f}_t \notin \mathcal{E}^* \}$.

Note that the last three states forms a partition of the sample space, and can potentially overlap with $B_t$. In addition, if $\gamma^* = 0$, this would imply that $\mathcal{C}^* = \varnothing$; in this case, states $S_t^+$ and $S_t^-$ will not ever be entered.

We first show a simple lemma that explains how non-steady states are related to having docile hypotheses in $\mathcal{F}_t$.

**Lemma 14.** *Suppose $\overline{B}_t$ happens. Then, the event $S_t^-$ implies that $\exists f \in \mathcal{D}^* . f \in \mathcal{F}_t$.*

*Proof.* Recall that $S_t^- = \left\{ \overline{f}_t \in \mathcal{E}^* \vee \widetilde{\mathcal{F}}_t \not\subseteq \mathcal{C}^* \right\} = \left\{ \overline{f}_t \in \mathcal{D}^* \right\} \cup \left\{ \overline{f}_t \in \mathcal{C}^* \right\} \cup \left\{ \overline{f}_t \in \mathcal{E}^*, \widetilde{\mathcal{F}}_t \not\subseteq \mathcal{C}^* \right\}$. In addition, $\overline{B}_t$ gives that $f^*$ is in $\mathcal{F}_t$. These imply one of the following:

1. $\overline{f}_t \in \mathcal{D}^*$; in this case, we are done.
2. $\overline{f}_t \in \mathcal{C}^*$. We first note that in this case, by the definition of $\mathcal{C}^*$ along with the unique best arm assumption (defined in Section 1), we have $\forall f \in \mathcal{C}^*, \mu^*(f) > \mu^*(f^*) = \mu^*$; this implies that $\overline{\mu}_t = \mu^*(\overline{f}_t) > \mu^*$. We consider two further subcases:

    (a) If $\widetilde{a}_t \neq a^*$, then by the definition of $(\overline{a}_t, \overline{\mu}_t) = \arg\min_{(a,\mu) \in \mathcal{B}_t : a \neq \widetilde{a}_t} \mu$, the range of $(a, \mu)$'s in the minimum includes $(a^*, \mu^*)$, we have $\mu^*(\overline{f}_t) = \overline{\mu}_t \leq \mu^*$. This contradicts with our premise that $\overline{\mu}_t > \mu^*$.

    (b) If $\widetilde{a}_t = a^*$, then consider any hypothesis $f_0 \in \widetilde{\mathcal{F}}_t$. We have
    $$f_0(a^*) = \widetilde{\mu}_t \geq \overline{\mu}_t > \mu^* = f^*(a^*),$$
    implying that $f_0 \in \mathcal{D}^*$.

3. $\overline{f}_t \in \mathcal{E}^* \wedge \widetilde{\mathcal{F}}_t \not\subseteq \mathcal{C}^*$. In this case, there must exist an element $f_0 \in \widetilde{\mathcal{F}}_t$ such that $f_0 \in \mathcal{D}^*$ or $f_0 \in \mathcal{E}^*$. We claim that the latter cannot happen. To see why, by the definition of $\widetilde{\mathcal{F}}_t$, it must be true that $\overline{\mathcal{F}}_t$ and $\widetilde{\mathcal{F}}_t$ belong to two different equivalence classes induced by relationship $\sim$. In addition, as $\overline{f}_t \in \mathcal{E}^*$, $\overline{\mathcal{F}}_t$ is a subset of the equivalence class $\mathcal{E}^*$. This implies that $\widetilde{\mathcal{F}}_t \cap \mathcal{E}^* = \varnothing$. Therefore, $f_0 \in \mathcal{D}^*$ must hold.

In summary, in all cases, we have that there exists some $f$ in $\mathcal{F}_t$ such that $f \in \mathcal{D}^*$. $\square$

We will bound the expected regret of CROP by a case-by-case analysis on the combination of observable and hidden states at each time step:

$$
\begin{aligned}
\mathbb{E} \operatorname{Reg}_n &= \mathbb{E} \sum_a \Delta_a \sum_{t=1}^n \mathbb{1}\{a_t = a\} \\
&\leq \mathbb{E} \sum_a \Delta_a \sum_{t=1}^n \mathbb{1}\{a_t = a, B_t\} + \mathbb{E} \sum_a \Delta_a \sum_{t=1}^n \mathbb{1}\{a_t = a, \overline{B}_t, \mathrm{Ex}_t\} \\
&\quad + \underbrace{\mathbb{E} \sum_a \Delta_a \sum_{t=1}^n \mathbb{1}\{a_t = a, \overline{B}_t, \overline{\mathrm{Ex}}_t, S_t^-\}}_{(\mathrm{Z1})} + \underbrace{\mathbb{E} \sum_a \Delta_a \sum_{t=1}^n \mathbb{1}\{a_t = a, \overline{B}_t, \overline{\mathrm{Ex}}_t, \overline{S_t^-}\}}_{(\mathrm{Z2})}
\end{aligned}
\tag{14}
$$

Note that the first two terms are easy to bound. First,

$$
\begin{aligned}
\mathbb{E} \sum_a \Delta_a \sum_{t=1}^n \mathbb{1}\{a_t = a\} &\leq \Delta_{\max} \mathbb{E} \sum_{t=1}^n \mathbb{1}\{B_t\} \\
&\leq \Delta_{\max} \sum_{t=1}^n \mathbb{P}\left( L_{t-1}(f^*) - \min_{f \in \mathcal{F}} L_{t-1}(f) > \beta_t \right) \\
&\leq \Delta_{\max} \left( 1 + |\mathcal{F}| \sum_{t=2}^n \frac{1}{zt(\log_2 t)^2} \right) \leq 3 \Delta_{\max} .
\end{aligned}
$$

where the second to last inequality is from Lemma 4(i), and the last inequality is by the values of $z$ and $\alpha$ stated in Theorem 3, and the elementary fact that $\sum_{t=2}^\infty \frac{1}{t(\log_2 t)^2} \leq \frac{1}{2} + \int_2^\infty \frac{1}{t(\log_2 t)^2} dt \leq 2$.

Second, if $\overline{B}_t$ and $\mathrm{Ex}_t$ happens, $a_t = a^*(\mathcal{F}_t) = a^*$. Therefore,

$$
\mathbb{E} \sum_a \Delta_a \sum_{t=1}^n \mathbb{1}\{a_t = a, \overline{B}_t, \mathrm{Ex}_t\} = \mathbb{E} \Delta_{a^*} \sum_{t=1}^n \mathbb{1}\{a_t = a^*, \overline{B}_t, \mathrm{Ex}_t\} = 0.
$$

To bound the third term (Z1), we use Lemma 18 in Section C.5.

For (Z2), we decompose (Z2) to a few more sub-terms. We first have the following claim:

**Claim 1.** *If $S_t^-$ does not happen, then $\mathrm{Fb}_t$ does not happen, i.e. either $\mathrm{Cf}_t$ or $\mathrm{Fs}_t$ happens.*

*Proof.* If $S_t^-$ does not happen, then we have $\overline{f}_t \in \mathcal{E}^*$ and $\widetilde{\mathcal{F}}_t \subseteq \mathcal{C}^*$ both hold. These imply that $\mathcal{C}^* = \mathcal{C}(\overline{f}_t)$, and consequently, $\widetilde{\mathcal{F}}_t \subseteq \mathcal{C}(\overline{f}_t)$. But this would imply that the condition of line 13 is satisfied and the state $\mathrm{Fb}_t$ will not be entered. $\qquad\square$

The above claim indicates that (Z2) can be bounded by:

$$(\text{Z2}) = \mathbb{E}\sum_a \Delta_a \sum_{t=1}^n \mathbb{1}\{a_t = a, \mathrm{Fs}_t, \overline{S_t^-}\} + \mathbb{E}\sum_a \Delta_a \sum_{t=1}^n \mathbb{1}\{a_t = a, \mathrm{Cf}_t, \overline{S_t^-}\}$$

$$= \underbrace{\mathbb{E}\sum_a \Delta_a \sum_{t=1}^n \mathbb{1}\{a_t = a, \mathrm{Fs}_t, S_t^+\}}_{(\text{Z2-a})} + \underbrace{\mathbb{E}\sum_a \Delta_a \sum_{t=1}^n \mathbb{1}\{a_t = a, \mathrm{Cf}_t, \overline{S_t^-}\}}_{(\text{Z2-b})}$$

$$+ \underbrace{\mathbb{E}\sum_a \Delta_a \sum_{t=1}^n \mathbb{1}\{a_t = a, \mathrm{Fs}_t, S_t^0\}}_{(\text{Z2-c})}.$$

Now, using Lemmas 15, 16, and 17 in Section C.5, we get

$$(\text{Z2}) \le P_1 \cdot \left(\ln n + 8\sqrt{\ln n \ln|\mathcal{F}|} + 2\ln\ln n + 23\ln|\mathcal{F}|\right) + 2K_\psi \Delta_{\max} + 64 P_2 \cdot \left(\ln(\ln(n)) + \ln(|\mathcal{F}|)\right) + 2K_\psi \Delta_{\max}$$

$$+ 200 \cdot \left(\sum_a \Delta_a \gamma_a(\mathcal{E}^*)\right) \ln|\mathcal{F}| + 5K_\psi \Delta_{\max}$$

$$\le P_1 \ln n + O\left(P_1\left(\sqrt{\ln n \ln|\mathcal{F}|} + \ln\ln n + \ln|\mathcal{F}|\right) + P_2 \ln(\ln n) + P_3 \cdot \ln|\mathcal{F}| + K_\psi \Delta_{\max}\right)$$

where the second inequality is by algebra, and the fact that $P_3 \ge P_1 \vee P_2 \vee (\sum_a \Delta_a \gamma_a(\mathcal{E}^*))$, which in turn is from the constraint in (4) we have $\psi_a(f) \ge \phi_a(f) \vee \gamma_a(f)$ for all $a \in \mathcal{A}$ and $f \in \mathcal{F}$.

Combining the above bound on (Z2) with Lemma 18 and Equation (14), we can bound the regret of CROP as follows:

$$\mathrm{Reg}_n \le (\text{Z1}) + (\text{Z2}) + 3\Delta_{\max}$$

$$\le P_1 \ln n + 3\Delta_{\max} + O\left(P_3(\ln|\mathcal{F}| + \ln(Q_1)) + K_\psi \Delta_{\max}\right)$$

$$+ O\left(P_1\left(\sqrt{\ln n \ln|\mathcal{F}|} + \ln\ln n + \ln|\mathcal{F}|\right)\right)$$

$$= P_1 \ln n + O\left(P_1\left(\sqrt{\ln n \ln|\mathcal{F}|} + \ln\ln n + \ln|\mathcal{F}|\right) + P_2 \ln(\ln(n)) + P_3\left(\ln(|\mathcal{F}|) + \ln(Q_1)\right) + K_\psi \Delta_{\max}\right).$$

If $\gamma^* = 0$, we have $P_1 = \sum_{a \in \mathcal{A}} \Delta_a \gamma_a^* = 0$. In addition, we must have $\mathcal{C}^* = \varnothing$, implying that for all $f \in \mathcal{E}^*$, $\gamma^*(f) = 0$. This in turn implies that for all $f, g$ in $\mathcal{E}^*$, $f \propto g$ is trivially true, and consequently $\phi(f) = 0$ for all $f \in \mathcal{E}^*$. Therefore, $\phi(\mathcal{E}^*) = 0$ and $P_2 = \sum_{a \in \mathcal{A}} \Delta_a \phi_a(\mathcal{E}^*) = 0$. The proof of Theorem 1 is complete. $\qquad\square$

## C.5 Bounding the regret in each individual case

### C.5.1 Bounding (Z2-a).

Recall that we use the shortcut $\gamma^* := \gamma(f^*)$. In addition, we have defined $P_1 = \sum_a \Delta_a \gamma_a^*$ and $\mathcal{A}_{\gamma^*} = \{a \in \mathcal{A} : \gamma_a^* \ne 0\}$. Note $|\mathcal{A}_{\gamma^*}| \le K_\psi$.

**Lemma 15.**

$$\mathbb{E}\sum_a \Delta_a \sum_{t=1}^n \mathbb{1}\{a_t = a, \mathrm{Fs}_t, S_t^+\} \le P_1\left(\ln n + 8\sqrt{\ln n \ln|\mathcal{F}|} + 2\ln\ln n + 23\ln|\mathcal{F}|\right) + 3|\mathcal{A}_{\gamma^*}|\Delta_{\max}.$$

*Proof.* By linearity of expectation,

$$\mathbb{E}\sum_a \Delta_a \sum_{t=1}^n \mathbb{1}\{a_t = a, \mathrm{Fs}_t, S_t^+\} = \sum_a \Delta_a \mathbb{E}\sum_{t=1}^n \mathbb{1}\{a_t = a, \mathrm{Fs}_t, S_t^+\}. \tag{15}$$

By Lemma 11, $\{a_t = a, \mathrm{Fs}_t, S_t^+\}$ will happen only for those $a$'s in $\mathcal{A}_{\gamma^*}$; thus only the arms in $\mathcal{A}_{\gamma^*}$ will contribute to the sum, which we focus on, hereafter.

With foresight, we pick $q_{1,a} = \left\lceil \frac{\beta_n}{2\sigma^2} \cdot \gamma_a^* \right\rceil$; clearly $q_{1,a} \le \frac{\beta_n}{2\sigma^2} \cdot \gamma_a^* + 1 \le \gamma_a^* \ln(|\mathcal{F}| n (\ln n)^2) + 1$.

Then,

$$\mathbb{E}\sum_{t=1}^n \mathbb{1}\{a_t = a, \mathrm{Fs}_t, S_t^+\}$$

$$\le q_{1,a} + 1 + \sum_{m=1}^\infty \mathbb{P}\{\exists t \,.\, a_t = a, \mathrm{Fs}_t, S_t^+, T_a(t-1) \ge q_{1,a} + m\}$$

$$= q_{1,a} + 1 + \sum_{m=1}^\infty \mathbb{P}\{\exists t \,.\, a_t = a, \mathrm{Fs}_t, S_t^+, T_a(t-1) \ge \gamma_a^* \cdot \frac{(q_{1,a} + m)}{\gamma_a^*}\}$$

$$\le q_{1,a} + 1 + \sum_{m=1}^\infty \min\left(1, |\mathcal{F}| \exp\left(-\frac{\gamma_a^*}{4(q_{1,a}+m)} \cdot \left(\frac{q_{1,a}+m}{\gamma_a^*} - \frac{\beta_n}{2\sigma^2}\right)^2\right)\right)$$

$$\le q_{1,a} + 1 + \sum_{m=1}^\infty \min\left(1, |\mathcal{F}| \exp\left(-\frac{m^2}{4(q_{1,a}+m)\gamma_a^*}\right)\right),$$

where the first inequality is from Lemma 8; the first equality is by decomposing $q_{1,a}+m = \gamma_a^* \cdot \frac{(q_{1,a}+m)}{\gamma_a^*}$; the second inequality is from Lemma 13(i); the third inequality is from the fact that $q_{1,a} = \left\lceil \frac{\beta_n}{2\sigma^2} \cdot \gamma_a^* \right\rceil \ge \frac{\beta_n}{2\sigma^2} \cdot \gamma_a^*$.

We now bound the last term of the last expression; observe that $\exp\left(-\frac{m^2}{4(q_{1,a}+m)\gamma_a^*}\right)$ is at most $\max\left(\exp\left(-\frac{m^2}{8q_{1,a})\gamma_a^*}\right), \exp\left(-\frac{m}{8\gamma_a^*}\right)\right)$. Therefore,

$$\sum_{m=1}^\infty \min\left(1, |\mathcal{F}| \exp\left(-\frac{m^2}{4(q_{1,a}+m)\gamma_a^*}\right)\right)$$

$$\le \sum_{m=1}^\infty \min\left(1, |\mathcal{F}| \max\left(\exp\left(-\frac{m^2}{8q_{1,a}\gamma_a^*}\right), \exp\left(-\frac{m}{8\gamma_a^*}\right)\right)\right)$$

$$\le \sum_{m=1}^\infty \min\left(1, |\mathcal{F}| \exp\left(-\frac{m^2}{8q_{1,a}\gamma_a^*}\right)\right) + \sum_{m=1}^\infty \min\left(1, |\mathcal{F}| \exp\left(-\frac{m}{8\gamma_a^*}\right)\right)$$

$$\le \int_0^\infty \min\left(1, |\mathcal{F}| \exp\left(-\frac{z^2}{8q_{1,a}\gamma_a^*}\right)\right) dz + \int_0^\infty \min\left(1, |\mathcal{F}| \exp\left(-\frac{z}{8\gamma_a^*}\right)\right) dz.$$

For the first integral, consider the boundary of its two segments $z_1 = \sqrt{8q_{1,a}\gamma_a^* \ln|\mathcal{F}|}$; it can then be simplified as

$$\int_0^\infty \min\left(1, |\mathcal{F}| \exp\left(-\frac{z^2}{8q_{1,a}\gamma_a^*}\right)\right) dz$$

$$= z_1 + \int_{z_1}^\infty |\mathcal{F}| \exp\left(-\frac{z^2}{8q_{1,a}\gamma_a^*}\right) dz$$

$$\le z_1 + \int_{z_1}^\infty |\mathcal{F}| \exp\left(-\frac{z^2}{8q_{1,a}\gamma_a^*}\right) \frac{z}{4q_{1,a}\gamma_a^*} dz \cdot \frac{4q_{1,a}\gamma_a^*}{z_1}$$

$$= z_1 + (-|\mathcal{F}|) \exp\left(-\frac{z^2}{8q_{1,a}\gamma_a^*}\right)\bigg|_{z_1}^{\infty} \cdot \frac{4q_{1,a}\gamma_a^*}{z_1}$$

$$\leq 2z_1.$$

For the second integral, consider the boundary between its two segments $z_2 = 8\gamma_a^* \ln|\mathcal{F}|$; it can then be simplified as

$$\int_0^{\infty} \min\left(1, |\mathcal{F}| \exp\left(-\frac{z}{8\gamma_a^*}\right) dz\right)$$

$$= z_2 + \int_{z_2}^{\infty} |\mathcal{F}| \exp\left(-\frac{z}{8\gamma_a^*}\right) dz$$

$$= z_2 + (-8\gamma_a^*)|\mathcal{F}| \exp\left(-\frac{z}{8\gamma_a^*}\right)\bigg|_{z_0}^{\infty}$$

$$\leq 2z_2.$$

In summary, we have shown that for all $a$,

$$\mathbb{E}\sum_{t=1}^{n} \mathbb{1}\{a_t = a, \mathrm{Fs}_t, S_t^+\} \leq q_{1,a} + 1 + 2z_1 + 2z_2$$

$$\leq q_{1,a} + 1 + \sqrt{32(q_{1,a}+1)\gamma_a^\star \ln|\mathcal{F}|} + 8\gamma_a^* \ln|\mathcal{F}|$$

$$\leq \gamma_a^* \ln(|\mathcal{F}| n(\ln n)^2) + 2 + \sqrt{32(\gamma_a^* \ln(|\mathcal{F}| n(\ln n)^2) + 1)\gamma_a^\star \ln|\mathcal{F}|} + 8\gamma_a^* \ln|\mathcal{F}|$$

$$\leq \gamma_a^*(\ln n + 8\sqrt{\ln n \ln|\mathcal{F}|} + 2\ln\ln n + 23\ln|\mathcal{F}|) + 3.$$

Continuing Equation (15), summing over all actions $a \in \mathcal{A}_{\gamma^*}$ with weight $\Delta_a$'s, we have

$$\sum_a \Delta_a \mathbb{E}\sum_{t=1}^{n} \mathbb{1}\{a_t = a, \mathrm{Fs}_t \wedge S_t^+\}$$

$$\leq \left(\sum_a \Delta_a \gamma_a^*\right) \cdot (\ln n + 8\sqrt{\ln n \ln|\mathcal{F}|} + 2\ln\ln n + 23\ln|\mathcal{F}|) + 3|\mathcal{A}_{\gamma^*}|\Delta_{\max}.$$

where the first inequality is from the fact that $\Delta_a \leq \Delta_{\max}$, and the second inequality is from the definition of $\beta_n$. The lemma follows from the definition of $P_1$. $\qquad\square$

### C.5.2 Bounding (Z2-b).

Recall that $P_2 = \sum_a \Delta_a \phi_a(\mathcal{E}^*)$ and $\mathcal{A}_{\phi(\mathcal{E}^*)} = \{a \in \mathcal{A} : \phi_a(\mathcal{E}^*) > 0\}$. Note $|\mathcal{A}_{\phi(\mathcal{E}^*)}| \leq K_\psi$.

**Lemma 16.**

$$\mathbb{E}\sum_a \Delta_a \sum_{t=1}^{n} \mathbb{1}\left\{a_t = a, \mathrm{Cf}_t, \overline{S_t^-}\right\} \leq 64 P_2 \cdot \left(\ln(\ln(n)) + \ln(|\mathcal{F}|)\right) + 2|\mathcal{A}_{\phi(\mathcal{E}^*)}|\Delta_{\max}.$$

*Proof.* First, we note that by Lemma 11, if $a$ is not in $\mathcal{A}_{\phi(\mathcal{E}^*)}$, it does not contribute to the sum, as $\mathrm{Cf}_t$ implies that only actions in $\mathcal{A}_{\phi(\mathcal{E}^*)}$ are taken with nonzero probability.

Next, by the linearity of expectation, we rewrite the expectation as follows:

$$\mathbb{E}\sum_a \Delta_a \sum_{t=1}^{n} \mathbb{1}\{a_t = a, \mathrm{Cf}_t, \overline{S_t^-}\} = \sum_{a \in \mathcal{A}_{\phi(\mathcal{E}^*)}} \Delta_a \mathbb{E}\left[\sum_{t=1}^{n} \mathbb{1}\{a_t = a, \mathrm{Cf}_t, \overline{S_t^-}\}\right]$$

For any $a \in \mathcal{A}_{\phi(\mathcal{E}^*)}$,

$$\mathbb{E}\sum_{t=1}^{n} \mathbb{1}\{a_t = a, \mathrm{Cf}_t, \overline{S_t^-}\} \leq 1 + \mathbb{E}\sum_{m=1}^{\infty} \mathbb{1}\{\exists t \leq n . a_t = a, T_a(t-1) \geq m, \mathrm{Cf}_t, \overline{S_t^-}\}$$

$$\leq 1 + \sum_{m=1}^{\infty} \mathbb{P}\left(\exists t \leq n \,.\, a_t = a, \mathrm{Cf}_t, T_a(t-1) \geq \frac{m}{\phi_a(\mathcal{E}^*)}\pi_{t,a}\right)$$

$$\leq 1 + \sum_{m=1}^{\infty} \min\left(1, \exp\left(-\frac{\frac{\sigma^2 m}{\phi_a(\mathcal{E}^*)} - 4\mathring{\beta}_n}{16\sigma^2}\right)\right)$$

where the first inequality is from Lemma 8 with $\tau = 1$; the second inequality is from the fact that if $T_a(t-1) \geq m$ and $\overline{S_t^-}$ happens, then $\overline{f}_t \in \mathcal{E}^*$, and therefore $T_a(t-1) \geq \frac{m}{\phi_a(\mathcal{E}^*)}\phi_a(\overline{f}_t) = \frac{m}{\phi_a(\mathcal{E}^*)}\pi_{t,a}$; the third inequality is from Lemma 13(iii) and $\mathbb{P}(A) \leq 1$ for any event $A$.

We remark that naively applying Lemma 7 instead of Lemma 8 as used in the case (Z2-a) (also used in the proofs of UCB [5] and UCB-S [27]), does not lead to the desired bound because of the aggressive confidence level of $\mathring{\mathcal{F}}_t$.

Denote by $N_m = \min\left(1, \exp\left(-\frac{\frac{\sigma^2 m}{\phi_a(\mathcal{E}^*)} - 4\mathring{\beta}_n}{16\sigma^2}\right)\right)$ and let $m_0 = \lceil 4\phi_a(\mathcal{E}^*)\frac{\mathring{\beta}_n}{\sigma^2}\rceil$.

For $m \leq m_0 - 1$, we use the fact that $N_m \leq 1$. For $m \geq m_0$, $\{N_m\}_{m \geq m_0}$ is a geometric progression with initial value $N_{m_0} \leq 1$ and common ratio $\exp(-\frac{1}{16\phi_a(\mathcal{E}^*)})$. This implies that

$$1 + \sum_{m=1}^{\infty} N_m \leq m_0 + \sum_{m=m_0}^{\infty} N_m \leq m_0 + \frac{1}{1 - \exp(-\frac{1}{16\phi_a(\mathcal{E}^*)})}$$

$$\leq 1 + 4\phi_a(\mathcal{E}^*)\frac{\mathring{\beta}_n}{\sigma^2} + (1 + 16\phi_a(\mathcal{E}^*))$$

$$\leq 2 + 64\phi_a(\mathcal{E}^*)\ln(|\mathcal{F}|\ln(n)).$$

<span style="color:green">Chicheng: With the new setting of $\mathring{\beta}_t$, the constant here can be improved.. But I think it is OK not to do that</span> where the first two inequalities are by algebra, the third inequality is from the definition of $m_0$ and the elementary fact that $\frac{1}{1-\exp(-1/x)} = 1 + \frac{1}{\exp(1/x)-1} \leq 1 + \frac{1}{((1/x)+1)-1} = 1 + x$ for $x > 0$; the last inequality is from the definition of $\mathring{\beta}_n$, $|\mathcal{F}| \geq 2$, $n \geq 2$ and algebra. Consequently,

$$\sum_{a \in \mathcal{A}_{\phi(\mathcal{E}^*)}} \Delta_a \mathbb{E}\left[\sum_{t=1}^{n} \mathbb{1}\left\{a_t = a, \mathrm{Cf}_t, \overline{S_t^-}\right\}\right]$$

$$\leq \sum_{a \in \mathcal{A}_{\phi(\mathcal{E}^*)}} \Delta_a\left(2 + 64\phi_a(\mathcal{E}^*)\ln(|\mathcal{F}|\ln(n))\right)$$

$$\leq 64\left(\sum_a \Delta_a\phi_a(\mathcal{E}^*)\right) \cdot \left(\ln(\ln(n)) + \ln(|\mathcal{F}|)\right) + 2|\mathcal{A}_{\phi(\mathcal{E}^*)}|\Delta_{\max},$$

where the second inequality uses the facts that $\Delta_a \leq \Delta_{\max}$ and algebra. The lemma follows from the definition of $P_2$. $\qquad\square$

### C.5.3 Bounding (Z2-c).

Recall that $\mathcal{A}_{\gamma(\mathcal{E}^*)} = \{a \in \mathcal{A} : \gamma_a(\mathcal{E}^*) > 0\}$. Note $|\mathcal{A}_{\gamma(\mathcal{E}^*)}| \leq K_\psi$.

**Lemma 17.**

$$\mathbb{E}\sum_a \Delta_a \sum_{t=1}^{n} \mathbb{1}\{a_t = a, \mathrm{Fs}_t, S_t^0\} \leq 200\left(\sum_a \Delta_a\gamma_a(\mathcal{E}^*)\right)\ln|\mathcal{F}| + 5|\mathcal{A}_{\gamma(\mathcal{E}^*)}|\Delta_{\max}.$$

*Proof.* First, we note that if $a$ is not in $\mathcal{A}_{\gamma(\mathcal{E}^*)}$, it does not contribute to the sum, as $\mathrm{Fs}_t$ implies that only actions in $\mathcal{A}_{\gamma(\mathcal{E}^*)}$ are taken with nonzero probability.

By linearity of expectation,

$$\mathbb{E}\sum_a \Delta_a \sum_{t=1}^n \mathbb{1}\{a_t = a, \mathrm{Fs}_t, S_t^0\} \le \sum_a \Delta_a \mathbb{E}\sum_{t=1}^n \mathbb{1}\{a_t = a, \mathrm{Fs}_t, S_t^0\}. \tag{16}$$

For every $a \in \mathcal{A}_{\gamma(\mathcal{E}^*)}$, we will upper bound $\mathbb{E}\sum_{t=1}^n \mathbb{1}\{a_t = a, \mathrm{Fs}_t, S_t^0\}$. To this end, we will upper bound $C_{a,n_0} := \mathbb{E}\sum_{t=n_0+1}^{2n_0} \mathbb{1}\{a_t = a, \mathrm{Fs}_t, S_t^0\}$, for every $n_0 \in \{2^k : k \in \{1, 2, \ldots\}\}$.

We first note that if $\mathrm{Fs}_t$ and $S_t^0$ both happen, then by the definition of $S_t^0$, $\overline{f}_t \sim f^*$, and $\mathring{\mathcal{F}}_t \subseteq \overline{\mathcal{F}}_t \subseteq \mathcal{E}^*$; we also have $\gamma(\overline{f}_t) \not\propto \gamma(f^*)$. In addition, by the definition of $\mathrm{Fs}_t$, for all $f, g \in \mathring{\mathcal{F}}_t$, $\gamma(f) \propto \gamma(g)$. Therefore, it must be the case that $f^* \notin \mathring{\mathcal{F}}_t$, implying that $\exists f \in \mathcal{E}^* \,.\, L_{t-1}(f^*) - L_{t-1}(f) > \mathring{\beta}_t$. We use this observation in the subsequent proof that we call "regret peeling".

With foresight, we pick $u_a = \lceil 5\gamma_a(\mathcal{E}^*)\frac{\beta_{2n_0}}{\sigma^2}\rceil$. We can bound $C_{a,n_0}$ as follows:

$$
\begin{aligned}
C_{a,n_0} &= \mathbb{E}\sum_{t=n_0+1}^{2n_0} \mathbb{1}\{a_t = a, \mathrm{Fs}_t, S_t^0\}\\[2mm]
&\le \mathbb{E}\sum_{t=n_0+1}^{2n_0} \mathbb{1}\{a_t = a, \mathrm{Fs}_t, \exists f \in \mathcal{E}^* \,.\, L_{t-1}(f^*) - L_{t-1}(f) > \mathring{\beta}_t\}\\[2mm]
&\le \mathbb{E}\sum_{t=n_0+1}^{2n_0} \mathbb{1}\{a_t = a, \mathrm{Fs}_t, \exists f \in \mathcal{E}^* \,.\, L_{t-1}(f^*) - L_{t-1}(f) > \mathring{\beta}_{n_0}\}\\[2mm]
&\le \mathbb{E}\,\mathbb{1}\left\{\exists s \in \mathbb{N}, f \in \mathcal{E}^* \,.\, L_s(f^*) - L_s(f) > \mathring{\beta}_{n_0}\right\} \cdot \sum_{t=n_0+1}^{2n_0} \mathbb{1}\left\{a_t = a, \mathrm{Fs}_t\right\}\\[2mm]
&\le \mathbb{E}\,\mathbb{1}\left\{\exists s \in \mathbb{N}, f \in \mathcal{E}^* \,.\, L_s(f^*) - L_s(f) > \mathring{\beta}_{n_0}\right\} \cdot \left(u_a + \sum_{t=n_0+1}^{2n_0} \mathbb{1}\left\{a_t = a, \mathrm{Fs}_t, T_a(t-1) \ge u_a\right\}\right)\\[2mm]
&\le \mathbb{P}\left(\exists s \in \mathbb{N}, f \in \mathcal{E}^* \,.\, L_s(f^*) - L_s(f) > \mathring{\beta}_{n_0}\right) \cdot u_a + \sum_{t=n_0+1}^{2n_0} \mathbb{P}\left(a_t = a, \mathrm{Fs}_t, T_a(t-1) \ge u_a\right)
\end{aligned}
$$

where the second inequality uses the basic fact that $\mathring{\beta}_t > \mathring{\beta}_{n_0}$ for all $t \ge n_0 + 1$; the third inequality is from the basic fact that $\mathbb{1}\{A, B\} = \mathbb{1}\{A\}\cdot\mathbb{1}\{B\}$ and the fact that for predicate $p$, $p(t)$ implies $\exists s \,.\, p(s)$.

The first term can be bounded by Lemma 4(i) and the union bound as follows:

$$
\begin{aligned}
&\mathbb{P}\left(\exists s \in \mathbb{N}, f \in \mathcal{E}^* \,.\, L_s(f^*) - L_s(f) > \mathring{\beta}_{n_0}\right) \cdot u_a\\[2mm]
&\le |\mathcal{F}|\exp\left(-\frac{\mathring{\beta}_{n_0}}{2\sigma^2}\right) \cdot u_a\\[2mm]
&\le \frac{1}{(\log_2 n_0)^3} + 20\gamma_a(\mathcal{E}^*)\cdot\left(\frac{\ln|\mathcal{F}|}{(\log_2 n_0)^3} + \frac{2}{(\log_2 n_0)^2}\right).
\end{aligned}
$$

where the last inequality uses $n_0 \ge 2$ and $\frac{\ln(2n_0\log(2n_0)^2)}{(\log_2(n_0))} \le 3 \cdot \frac{\ln 2 + \ln n_0}{\log_2 n_0} \le 5$.

**Remark 2.** We remark that the inequality above is the one that reflects our intuition that, even if we track a wrong $\gamma(f)$ with $f \in \mathcal{E}^*$ and suffer regret like $\sum_a \Delta_a \gamma_a(f)\ln(t)$ up to time $t$ (that can be much larger than $\sum_a \Delta_a \gamma_a^*\ln(t)$), such an event happens with small enough probability like $O(\frac{1}{\ln(t)})$. Therefore, *in expectation*, this event contributes to the regret only as a finite term w.r.t. $n$. This intuition is manifested in the proof in a bit more complicated way, unfortunately, because the algorithm is designed to enjoy an anytime regret bound rather than the fixed-budget setting. Specifically, the failure rate of the confidence set $\mathring{\mathcal{F}}_t$ changes over time, and we use the common technique called "peeling device" from concentration of measure to deal with it. If we knew the time horizon $n$, then one can set $\mathring{\beta}_t = 4\sigma^2\ln(|\mathcal{F}|\ln n)$ for all $t$ to obtain the same guarantee.

Meanwhile, each subterm in the second term can be bounded using Lemma 13(ii) as follows:

$$\mathbb{P}\left(a_t = a, \mathrm{Fs}_t, T_a(t-1) \geq u_a\right) \leq \mathbb{P}\left(a_t = a, \mathrm{Fs}_t, T_a(t-1) \geq \frac{u_a}{\gamma_a(\mathcal{E}^*)}\pi_{t,a}\right)$$

$$\leq |\mathcal{F}| \cdot \exp\left(-\frac{\sigma^2 \frac{u_a}{\gamma_a(\mathcal{E}^*)} - \beta_t}{4\sigma^2}\right) \leq \frac{1}{(2n_0)^2}.$$

where the last inequality is from the definition of $u_a$ (which in turn implies that $\sigma^2 \frac{u_a}{\gamma_a(\mathcal{E}^*)} \geq 5\beta_{2n_0}$) and $\beta_{2n_0} \geq \beta_t$, and $\exp(-\frac{\beta_{n_0}}{\sigma^2}) \leq \frac{1}{(2n_0)^2}$. This implies that

$$\sum_{t=n_0+1}^{2n_0} \mathbb{P}\left(a_t = a, \mathrm{Fs}_t, T_a(t-1) \geq u_a\right) \leq \frac{1}{4n_0}.$$

In summary, we have

$$C_{a,n_0} \leq \left(\frac{1}{(\log_2 n_0)^3} + \frac{1}{4n_0}\right) + 20\gamma_a(\mathcal{E}^*) \cdot \left(\frac{\ln|\mathcal{F}|}{(\log_2 n_0)^3} + \frac{5}{(\log_2 n_0)^2}\right).$$

Now, we can upper bound $\mathbb{E} \sum_{t=1}^n \mathbb{1}\{a_t = a, \mathrm{Fs}_t, S_t^0\}$ as follows:

$$\mathbb{E}\sum_{t=1}^n \mathbb{1}\{a_t = a, \mathrm{Fs}_t \wedge S_t^0\} \leq 2 + \sum_{k=1}^\infty C_{a,2^k}$$

$$\leq 2 + \sum_{k=1}^\infty \left(\frac{1}{k^3} + \frac{1}{2^k}\right) + 16\gamma_a(\mathcal{E}^*) \cdot \left(\sum_{k=1}^\infty \frac{\ln|\mathcal{F}|}{k^3} + \frac{2}{k^2}\right)$$

$$\leq 5 + 200\gamma_a(\mathcal{E}^*) \cdot \ln|\mathcal{F}|,$$

where the last inequality is by algebra and $1 \leq 2\ln|\mathcal{F}|$ due to $|\mathcal{F}| \geq 2$.

Using the bound above, continuing Equation (16), we have

$$\sum_{a \in \mathcal{A}_{\gamma(\mathcal{E}^*)}} \Delta_a \mathbb{E}\sum_{t=1}^n \mathbb{1}\{a_t = a, \mathrm{Fs}_t \wedge S_t^0\} \leq \sum_{a \in \mathcal{A}_{\gamma(\mathcal{E}^*)}} \Delta_a(5 + 200\gamma_a(\mathcal{E}^*) \cdot \ln|\mathcal{F}|)$$

$$\leq 200\left(\sum_a \Delta_a \gamma_a(\mathcal{E}^*)\right)\ln|\mathcal{F}| + 5|\mathcal{A}_{\gamma(\mathcal{E}^*)}|\Delta_{\max},$$

where the last inequality uses $\Delta_a \leq \Delta_{\max}$ for all $a \in \mathcal{A}$. □

### C.5.4 Bounding (Z1)

Recall that $P_3 = \sum_a \Delta_a \psi_a(\mathcal{F})$, $\Lambda_{\min} = \min_{f \in \mathcal{D}^*} \frac{|f(a^*) - \mu^*|}{\sigma}$ is the smallest information gap, and $Q_1 = \Lambda_{\min}^{-2} + K_\psi(1 + \max_a \psi_a(\mathcal{F}))$.

**Lemma 18.**

$$\mathbb{E}\sum_a \Delta_a \sum_{t=1}^n \mathbb{1}\{a_t = a, \overline{B}_t, \overline{\mathrm{Ex}_t}, S_t^-\} \leq O\left(P_3(\ln|\mathcal{F}| + \ln(Q_1)) + K_\psi \Delta_{\max}\right). \qquad (17)$$

*Proof.* Recall that $\alpha = 2$. With foresight, define

$$\tau = \max\left\{t \in \mathbb{N}_+ : (t < 8) \ \vee \ \left(\frac{t}{2} < 1 + \frac{6}{\Lambda_{\min}^2 \sigma^2}\beta_t\right) \ \vee \ \left(\frac{t}{4K_\psi} < 1 + 12\frac{\beta_t}{\sigma^2}(\max_a \psi_a(\mathcal{F}))\right)\right\}. \tag{18}$$

We upper bound the LHS of Equation (17) with three terms:

$$\sum_a \Delta_a \mathbb{E}\sum_{t=1}^n \mathbb{1}\{a_t = a, \overline{B}_t, \overline{\mathrm{Ex}_t}, S_t^-\}$$

$$\leq \mathbb{E}\sum_a \Delta_a \sum_{t=1}^{\tau} \mathbb{1}\{a_t = a, \overline{B}_t, \overline{\mathrm{Ex}_t}, S_t^-\} + \mathbb{E}\sum_{t=\tau+1}^{n}\sum_a \Delta_a \mathbb{1}\{a_t = a, \overline{B}_t, \overline{\mathrm{Ex}_t}, S_t^-, T_{a^*}(t-1) \geq \frac{t}{2} - 1\}$$

$$+ \mathbb{E}\sum_{t=\tau+1}^{n}\sum_a \Delta_a \mathbb{1}\{a_t = a, \overline{B}_t, \overline{\mathrm{Ex}_t}, S_t^-, T_{a^*}(t-1) < \frac{t}{2} - 1\}$$

$$\leq \underbrace{\sum_a \Delta_a \mathbb{E}\sum_{t=1}^{\tau} \mathbb{1}\{a_t = a, \overline{B}_t, \overline{\mathrm{Ex}_t}, S_t^-\}}_{\text{(Z1-a)}} + \underbrace{\Delta_{\max}\sum_{t=\tau+1}^{n} \mathbb{P}\left(\overline{B}_t, \overline{\mathrm{Ex}_t}, S_t^-, T_{a^*}(t-1) \geq \frac{t}{2} - 1\right)}_{\text{(Z1-b)}}$$

$$+ \underbrace{\Delta_{\max}\sum_{t=\tau+1}^{n} \mathbb{P}\left(T_{a^*}(t-1) < \frac{t}{2} - 1\right)}_{\text{(Z1-c)}}$$

where the first inequality is by algebra; the second inequality uses the fact that $\Delta_a \leq \Delta_{\max}$ for all $a$, and linearity of expectation.

We bound each term respectively.

**Bounding (Z1-a).** Recall $\mathcal{A}_\psi = \{a \in \mathcal{A} : \psi(\mathcal{F}) > 0\}$. Define $\mathcal{A}'_\psi = \mathcal{A}_\psi \setminus \{a^*\}$. First, we note that if $a$ is not in $\mathcal{A}'_\psi$, by Lemma 11, it does not contribute to the sum. This is because, if $\overline{B}_t$ happens, the only arms being pulled is either $a^*$ or from $\mathcal{A}'_\psi$ (and $\Delta_{a^*} = 0$).

With the choice of $q_{3,a} = \lceil 12\frac{\beta_\tau}{\sigma^2}\psi_a(\mathcal{F})\rceil$, we have

$$\mathbb{E}\sum_{t=1}^{\tau} \mathbb{1}\{a_t = a, \overline{B}_t, \overline{\mathrm{Ex}_t}, S_t^-\}$$

$$\leq \mathbb{E}\sum_{t=1}^{\tau} \mathbb{1}\{a_t = a, \overline{\mathrm{Ex}_t}, S_t^-\}$$

$$\leq q_{3,a} + \sum_{t=1}^{\tau} \mathbb{P}\left(a_t = a, \overline{\mathrm{Ex}_t}, T_a(t-1) \geq q_{3,a}\right) \qquad (\because \text{ Lemma 7})$$

$$\leq q_{3,a} + \sum_{t=1}^{\tau} \mathbb{P}\left(a_t = a, \overline{\mathrm{Ex}_t}, T_a(t-1) \geq 12\frac{\beta_\tau}{\sigma^2}\pi_{t,a}\right) \qquad (\because q_{3,a} \geq 12\frac{\beta_\tau}{\sigma^2}\psi_a(\mathcal{F}))$$

$$\leq q_{3,a} + \sum_{t=1}^{\tau} |\mathcal{F}|\exp\left(-\frac{12\beta_\tau - 4\beta_t}{16\sigma^2}\right) \qquad (\because \text{ Lemma 13(ii)})$$

$$\leq 12\frac{\beta_\tau}{\sigma^2}\psi_a(\mathcal{F}) + 1 + \sum_{t=1}^{\tau} \frac{1}{\tau(\log_2 \tau)^2} \qquad (\because \beta_t \leq \beta_\tau, \text{ the definition of } \beta_\tau)$$

$$\leq 24\psi_a(\mathcal{F})(\ln|\mathcal{F}| + \ln\tau) + 3 \qquad (\because \sum_{t=1}^{\tau}\frac{1}{\tau(\log_2 \tau)^2} = \frac{1}{(\log_2 \tau)^2} \text{ and } \tau \geq 8)$$

$$= O(\psi_a(\mathcal{F}) \cdot (\ln|\mathcal{F}| + \ln Q_1) + 1),$$

where the last inequality is from Lemma 20 below where we show $\ln\tau = O(\ln(Q_1) + \ln\ln|\mathcal{F}|)$.

Summing over all $a \in \mathcal{A}'_\psi$, we have

$$\sum_a \Delta_a \mathbb{E}\sum_{t=1}^{\tau} \mathbb{1}\{a_t = a, \overline{B}_t, \overline{\mathrm{Ex}_t}, S_t^-\} = O\left(\left(\sum_a \Delta_a\psi_a(\mathcal{F})\right)(\ln|\mathcal{F}| + \ln(Q_1)) + K_\psi\Delta_{\max}\right).$$

**Bounding (Z1-b).** In subsequent derivations, we denote by $X_t = \frac{6}{\Lambda_{\min}^2}\frac{\beta_t}{\sigma^2}$.

$$\sum_{t=\tau+1}^{n} \mathbb{P}\left(\overline{B}_t, \overline{\mathrm{Ex}_t}, S_t^-, T_{a^*}(t-1) \geq \frac{t}{2} - 1\right)$$

$$\leq \sum_{t=\tau+1}^{n} \mathbb{P}\left(\overline{B}_t, \overline{\mathrm{Ex}_t}, S_t^-, T_{a^*}(t-1) \geq X_t\right)$$

$$\leq \sum_{t=\tau+1}^{n} \mathbb{P}\left(T_{a^*}(t-1) \geq X_t, \exists f \in \mathcal{D}^* . L_{t-1}(f) - L_{t-1}(f^*) \leq \beta_t\right)$$

$$\leq \sum_{t=\tau+1}^{n} \mathbb{P}\left(\exists f \in \mathcal{D}^* . \mathrm{IC}^*(f, T(t-1)) \geq 3\frac{\beta_t}{\sigma^2}, L_{t-1}(f) - L_{t-1}(f^*) \leq \beta_t\right)$$

$$\leq \sum_{t=\tau+1}^{n} |\mathcal{F}| \exp\left(-\frac{2\beta_t}{4\sigma^2}\right)$$

$$\leq \sum_{t=2}^{n} \frac{1}{t(\log_2 t)^2} \leq 2,$$

where the first inequality is from the definition of $\tau$: for every $t > \tau$, $\frac{t}{2} - 1 \geq X_t$; the second inequality is from Lemma 14; the third inequality is from the observation that $\mathrm{IC}^*(f, T(t-1)) \geq \frac{1}{2}T_{a^*}(t-1)\Lambda_{a^*}(f)^2 \geq \frac{1}{2}T_{a^*}(t-1)\Lambda_{\min}^2 \geq 3\frac{\beta_t}{\sigma^2}$; the fourth inequality is from Lemma 4(ii); the last two inequalities are by algebra.

**Bounding (Z1-c).** We first bound $\mathbb{P}\left(T_{a^*}(t-1) < \frac{t}{2} - 1\right)$ for each $t$. First, denote by $I = \left[\lfloor \frac{t}{4} \rfloor + 1, t - 1\right]$. In this notation, we claim that the following implication holds:

$$\left\{T_{a^*}(t-1) < \frac{t}{2} - 1\right\} \cap \left(\bigcap_{s \in I} \overline{B_s}\right) \subseteq \left\{\exists a \in \mathcal{A}'_\psi, s \in I . \overline{B_s}, a_s = a, T_a(s-1) \geq \frac{t}{4K_\psi} - 1\right\}. \quad (19)$$

Indeed, if $\bigcap_{s \in I} \overline{B_s}$ holds, then the chosen arm $a_s$ at time step $s$ must come from $\mathcal{A}_\psi = \mathcal{A}'_\psi \cup \{a^*\}$; the reason is as follows:

1. if $\mathrm{Ex}_s$, then $a_s = a^*$ is pulled;
2. otherwise, $a_s$ is drawn from $\pi_s$ which is supported on $\mathcal{A}_\psi$.

Throughout time interval $I$, we note that there are $\geq t - 1 - \lfloor \frac{t}{4} \rfloor \geq \frac{3}{4}t - 1$ time steps. Given the premise that $T_{a^*}(t-1) < \frac{t}{2} - 1$, the number of arm pulls of $a^*$ in $I$ must be $< \frac{t}{2} - 1$; this implies that the total number of arm pulls in $\mathcal{A}'_\psi$ in $I$ must be greater than $(\frac{3t}{4} - 1) - (\frac{t}{2} - 1) \geq \frac{t}{4}$. By pigeonhole's principle, there exists an arm $a_0 \in \mathcal{A}'_\psi$ such that the number of arm pulls of $a_0$ in time span $I$ is at least $\frac{t}{4K_\psi}$. Let $s$ be the last time step in $I$ when $a_0$ is pulled; therefore, we have $a_s = a_0, T_{a_0}(s-1) \geq \frac{t}{4K_\psi} - 1$, and $\overline{B_s}$ holding simultaneously, proving the above implication.

Equation (19) is equivalent to

$$\left\{T_{a^*}(t-1) < \frac{t}{2} - 1\right\} \cap \left(\bigcap_{s \in I} \overline{B_s}\right) \subseteq \bigcup_{s \in I}\left\{\overline{B_s}, \exists a \in \mathcal{A}'_\psi . a_s = a, T_a(s-1) \geq \frac{t}{4K_\psi} - 1\right\}.$$

Therefore, by the elementary fact that $\mathbb{P}(U) \leq P(V) + P(\overline{V} \cap U)$ and De Morgan's Law, we have

$$\mathbb{P}\left(T_{a^*}(t-1) < \frac{t}{2} - 1\right) \leq \mathbb{P}\left(\bigcup_{s \in I} B_s\right) + \mathbb{P}\left(\bigcup_{s \in I}\left\{\overline{B_s}, \exists a \in \mathcal{A}'_\psi . a_s = a, T_a(s-1) \geq \frac{t}{4K_\psi} - 1\right\}\right).$$

For the first term, by Lemma 6, we have

$$\mathbb{P}\left(\bigcup_{s \in I} B_s\right) \leq \mathbb{P}\left(\bigcup_{s \geq \lfloor \frac{t}{4} \rfloor + 1} B_s\right) \leq \frac{1}{\frac{t}{4}(\log_2 \frac{t}{4})^2} \leq \frac{36}{t(\log_2 t)^2},$$

where the last inequality uses the elementary fact that $\log_2 \frac{t}{4} \geq \frac{1}{3}\log_2 t$ for $t \geq \tau \geq 8$. For the second term, we have:

$$\mathbb{P}\left(\bigcup_{s \in I}\left\{\overline{B_s}, \exists a \in \mathcal{A}'_\psi . a_s = a, T_a(s-1) \geq \frac{t}{4K_\psi} - 1\right\}\right)$$

$$\leq \sum_{a \in \mathcal{A}'_\psi} \mathbb{P}\left(\exists s \in I . \overline{B_s}, a_s = a, T_a(s-1) \geq \frac{t}{4K_\psi} - 1\right)$$

$$\leq \sum_{a \in \mathcal{A}'_\psi} \mathbb{P}\left(\exists s \in I . \overline{\mathrm{Ex}_s}, a_s = a, T_a(s-1) \geq 12\frac{\beta_t}{\sigma^2} \cdot \psi_a(\mathcal{F})\right)$$

$$\leq \sum_{a \in \mathcal{A}'_\psi} \mathbb{P}\left(\exists s \in I . \overline{\mathrm{Ex}_s}, a_s = a, T_a(s-1) \geq 12\frac{\beta_t}{\sigma^2} \cdot \pi_{s,a}\right)$$

$$\leq K_\psi \cdot |\mathcal{F}| \cdot \frac{1}{zt^\alpha} = \frac{K_\psi}{t(\log_2 t)^2},$$

where the first inequality is by union bound; the second inequality is from the definition of $\tau$: for all $t \geq \tau + 1$, $\frac{t}{4K_\psi} - 1 \geq 12\frac{\beta_t}{\sigma^2}(\max_a \psi_a(\mathcal{F})) \geq 12\frac{\beta_t}{\sigma^2} \cdot \psi_a(\mathcal{F})$ and the fact that $\mathcal{A}'_\psi$ does not contain $a^*$; the third inequality is from the observation that $\psi_a(\mathcal{F}) \geq \psi_a(\overline{f}_s) \geq \pi_{s,a}$; the fourth inequality is from Lemma 13(ii) and the fact that $\beta_t = \exp\left(-\frac{\beta_t}{2\sigma^2}\right) \leq \frac{1}{|\mathcal{F}|t(\log_2 t)^2}$; the last inequality is by algebra.

To summarize,

$$\mathbb{P}\left(T_{a^*}(t-1) < \frac{t}{2} - 1\right) \leq \frac{K_\psi + 36}{t(\log_2 t)^2}.$$

Summing over all $t$'s, we get that

$$\Delta_{\max} \sum_{t=\tau+1}^{n} \mathbb{P}\left(T_{a^*}(t-1) < \frac{t}{2} - 1\right) \leq \Delta_{\max} \sum_{t=3}^{\infty} \frac{K_\psi + 36}{t(\log_2 t)^2} \leq 2(K_\psi + 36)\Delta_{\max}.$$

**Putting all together.** Combining the bounds on (Z1-a), (Z1-b), (Z1-c), we have

$$(\text{Z1}) \leq O\left(\left(\sum_a \Delta_a \psi_a(\mathcal{F})\right) \cdot \left(\ln|\mathcal{F}| + \ln(Q_1)\right) + K_\psi \Delta_{\max}\right).$$

Applying the definition of $P_3$ concludes the proof. $\qquad\square$

## C.6  Miscellaneous lemmas

**Lemma 19.** *Let $A, B > 0$. Then, $t < A + B\log(t) \implies \ln t < 2\ln(B + \sqrt{A})$.*

*Proof.* We use $\log(t) \leq \sqrt{t}$:

$$t < A + B\log(t)$$
$$\leq A + B\sqrt{t}$$
$$\implies \sqrt{t} < \frac{B + \sqrt{B^2 + 4A}}{2}$$

Taking logs on both sides,

$$\frac{1}{2}\log t = \log(\sqrt{t}) \leq \ln\left(\frac{B + \sqrt{B^2 + 4A}}{2}\right) \leq \ln(B + \sqrt{A}).$$

$\qquad\square$

**Lemma 20.** *Suppose $\tau$ is defined as in Equation (18). Then,*

$$\log(\tau) = O(\ln(Q_1) + \ln(\ln(|\mathcal{F}|))). \tag{20}$$

*where $Q_1$ is defined in Theorem 3.*

*Proof.* Let $R := \max_a \psi_a(\mathcal{F})$. The constraint set that defines $\tau$, according to Equation (18), can be rewritten as:

$$\{\tau \leq 8\} \cup \left\{(\tau \geq 9) \wedge \left(\frac{\tau}{2} < 1 + \frac{6}{\Lambda_{\min}^2} \cdot 2\ln(|\mathcal{F}|\tau(\log_2 \tau)^2)\right)\right\} \cup \left\{(\tau \geq 9) \wedge \left(\frac{\tau}{4K_\psi} < 1 + 12 \cdot 2R \cdot \ln(|\mathcal{F}|\tau(\log_2 \tau)^2)\right)\right\}$$

Observe that when $\tau \geq 9$, $(\log_2 \tau)^2 \leq \tau$. This implies that,

$$(\tau \geq 9) \wedge \left(\frac{\tau}{2} < 1 + \frac{6}{\Lambda_{\min}^2} \cdot 2\ln(|\mathcal{F}|\tau(\log_2 \tau)^2)\right) \implies \tau < 2 + \frac{64}{\Lambda_{\min}^2}\ln|\mathcal{F}| + \frac{64}{\Lambda_{\min}^2}\ln(\tau),$$

$$(\tau \geq 9) \wedge \left(\frac{\tau}{4K_\psi} < 1 + 12 \cdot 2R \cdot \ln(|\mathcal{F}|\tau(\log_2 \tau)^2)\right) \implies \tau < 4K_\psi + 256K_\psi R\ln|\mathcal{F}| + 256K_\psi R\ln(\tau).$$

By the definition of $\tau$, we have

$$\tau \leq \max\left\{8, 2 + \frac{64}{\Lambda_{\min}^2}\ln|\mathcal{F}| + \frac{64}{\Lambda_{\min}^2}\ln(\tau), \ \ 4K_\psi + 256K_\psi R\ln|\mathcal{F}| + 256K_\psi R\ln(\tau)\right\}$$

$$\leq \max\left\{2 + \frac{64}{\Lambda_{\min}^2}\ln|\mathcal{F}| + \frac{64}{\Lambda_{\min}^2}\ln(\tau), \ \ 8K_\psi, \ \ 512K_\psi R\ln|\mathcal{F}| + 512K_\psi R\ln(\tau)\right\}$$

where the second inequality is by $a + b \leq \max\{2a, 2b\}$. We can compactly write down $\tau \leq \max\{8K_\psi, \ A + B\ln(\tau)\}$ with

$$A = \max\left\{2 + \frac{64}{\Lambda_{\min}^2}\ln|\mathcal{F}|, \ \ 512K_\psi R\ln|\mathcal{F}|\right\}$$

$$B = \max\left\{\frac{64}{\Lambda_{\min}^2}, \ \ 512K_\psi R\right\}.$$

Then, by Lemma 19, we have

$$\ln\tau \leq \max(\ln 8K_\psi, 2\ln(B + \sqrt{A})) \leq \max(\ln 8K_\psi, 2\ln(A + B))$$

Let $\xi = \Lambda_{\min}^{-2} + K_\psi R$. Because $A = \Theta(1 + \xi\ln(|\mathcal{F}|))$, $B = \Theta(\xi)$, we have

$$\ln\tau \leq O(\ln K_\psi + \ln(1 + \xi\ln(|\mathcal{F}|))$$
$$= O(\ln(K_\psi + \xi\ln(|\mathcal{F}|))$$
$$= O(\ln(\Lambda_{\min}^{-2} + K_\psi(1 + R)) + \ln(\ln(|\mathcal{F}|))$$

where the first equality uses $\ln(a) + \ln(b) = 2\max(\ln(a), \ln(b)) \leq 2\ln(\max(a, b))$, and the second inequality is from the definition of $\xi$ and algebra. $\qquad\square$

## D  Lower bound

For our lower bound, we consider the following instance that resembles $\mathcal{H}^+$ from Figure 2.

**Example 1.** Let $\epsilon, \Lambda > 0$ and $r > 1$. Suppose $\epsilon$ is small enough to ensure that $f_2$ has the only informative arm of 3 and $f_3$ has the only informative arm of 4.

- $f_1 = (1, 1 + \epsilon, 0, 0)$
- $f_2 = (1, 1 - \epsilon, \Lambda, 0)$
- $f_3 = (1, 1 - \epsilon, \Lambda, r\Lambda)$

Let us denote by $\mathbb{E}_i T_j(n)$ the expected number of pulls of arm $j$ under the instance $f_i$. We state our lower bound result in the following theorem.

**Theorem 21.** *Consider Example 1. Assume the Gaussian noise model with $\sigma^2 = 1$. Suppose a bandit algorithm has $\mathbb{E}_1 T_1(n) = O(n^u)$ for some $u \in [0, 1)$, then, for sufficiently large $n$,*

$$\mathbb{E}_3 T_2(n) \vee \mathbb{E}_3 T_3(n) \vee \mathbb{E}_2 T_4(n) \geq \frac{6}{5}\frac{1}{r^2\Lambda^2}\ln\left(1 + \frac{(1 - u)\ln(n)}{48}\right) = \Omega(\ln(1 + (1 - u)\ln n)).$$

At first sight, intuition on why the statement is true is not obvious since, assuming $\mathbb{E}_3 T_2(n) \vee \mathbb{E}_3 T_3(n)$ is $O(\ln(\ln((1 - u)n)))$, somehow the fact that arm 1 is not pulled sufficiently under $f_1$ implies a lower bound on arm 4 under $f_2$. To explain this, let us consider the contraposition: If $\mathbb{E}_2 T_4(n) < O(\ln(\ln((1 - u)n)))$ for large enough $n$, then $T_1(n) = \Omega(n^u)$. What happens in a nutshell is as

follows. The fact that $\mathbb{E}_2 T_4(n)$ is not sufficient means that the algorithm cannot distinguish between $f_2$ and $f_3$ with probability approaching to 1. Thus, roughly speaking, the behavior of the algorithm under $f_2$ and $f_3$ must be very similar for most times. Together with the assumption on $\mathbb{E}_3 T_2(n)$ and $\mathbb{E}_3 T_3(n)$, the algorithms does not collect sufficient number of samples from arm 2, 3, and 4 under both $f_2$ and $f_3$. The implication is that $(i)$ the remaining pulls all go to arm 1 and $(ii)$ the algorithm cannot distinguish between $f_1$ from $\{f_2, f_3\}$ either for most times, so it collects a lot of samples from arm 1 even under $f_1$. The specific reason why $\ln(\ln(n))$ appears is quite technical.

This has an implication on forced sampling, as discussed in Section 5. That is, the contraposition of Theorem 21 implies that naïvely pulling $\Theta(\ln(\ln(n)))$ for each arm may lead to regret that is arbitrarily close to being linear, let alone being uniformly good or being around the asymptotic optimality!

*Proof.* Let $Y_1, \ldots, Y_4 > 0$ be some constants such that $Y_1 + Y_2 + Y_3 + Y_4 = n$, to be tuned later. Define $Y_{i:j}$ as $Y_i, Y_{i+1}, \ldots, Y_j$ for convenience. Denote by $\mathbb{E}_i$ and $\mathbb{P}_i$ the expectation and probability under $f_i$, respectively. Using divergence decomposition and Bretagnolle–Huber inequality [30, Lemma 15.1 and Theorem 14.2, respectively], we have

$$
\begin{aligned}
\frac{1}{2} \exp & \left( -\mathbb{E}_2 \left[ \sum_{t=1}^n \mathsf{KL}(f_2(a_t), f_3(a_t)) \right] \right) \\
&= \frac{1}{2} \exp\left( -\frac{r^2 \Lambda^2}{2} \mathbb{E}_2 T_4(n) \right) \\
&\leq \mathbb{P}_2(T_4(n) \geq Y_4) + \mathbb{P}_3(T_4(n) < Y_4) \\
&= \mathbb{P}_2(T_4(n) \geq Y_4) + \mathbb{P}_3(T_4(n) < Y_4, T_3(n) \geq Y_3) \\
&\qquad + \mathbb{P}_3(T_4(n) < Y_4, T_3(n) < Y_3, T_2(n) \geq Y_2) \\
&\qquad + \mathbb{P}_3(T_4(n) < Y_4, T_3(n) < Y_3, T_2(n) < Y_2, T_1(n) \geq Y_1)
\end{aligned}
$$

On the other hand,

$$
\begin{aligned}
\mathbb{E}_2 &T_4(n) + \mathbb{E}_3 T_3(n) + \mathbb{E}_3 T_2(n) \\
&\geq Y_4 \, \mathbb{P}_2(T_4(n) \geq Y_4) + Y_3 \, \mathbb{P}_3(T_3(n) \geq Y_3) + Y_2 \, \mathbb{P}_3(T_2(n) \geq Y_2) \\
&\geq \min\{Y_2, Y_3, Y_4\} \cdot \big( \mathbb{P}_2(T_4(n) \geq Y_4) + \mathbb{P}_3(T_3(n) \geq Y_3) + \mathbb{P}_3(T_2(n) \geq Y_2) \big)
\end{aligned}
$$

Together,

$$
\begin{aligned}
\frac{1}{2} \exp\left( -\frac{r^2 \Lambda^2}{2} \mathbb{E}_2 T_4(n) \right) \leq{}& \frac{1}{\min\{Y_{2:4}\}} \big( \mathbb{E}_2 T_4(n) + \mathbb{E}_3 T_3(n) + \mathbb{E}_3 T_2(n) \big) \\
&+ \underbrace{\mathbb{P}_3(T_4(n) < Y_4, T_3(n) < Y_3, T_2(n) < Y_2, T_1(n) \geq Y_1)}_{=:\ Q_n}
\end{aligned} \tag{21}
$$

The main effort is spent on bounding $Q_n$.

Recall that reward distribution is Gaussian with variance $\sigma^2 = 1$. Let $p_f(r_s \mid a_s)$ be the pdf of the reward distribution under $f^* = f$ when arm $a_s$ is pulled at time $s$.

We have the following anytime inequality. Denote by $\mathbb{P}_{f^*}(\cdot)$ be the probability of an event when $f^*$ is the ground truth. The following lemma states that under $f^*$ the empirical KL-divergence is not too far from the KL-divergence (that is controlled by expected arm pulls made by the algorithm) with high probability.

**Lemma 22.** *For every $\rho > 0$,*

$$
\mathbb{P}_{f^*}\left( B(f^*, f) := \left\{ \exists t \geq 1, \ \sum_{s=1}^t \ln \frac{p_{f^*}(r_s \mid a_s)}{p_f(r_s \mid a_s)} \geq (1 + \rho) \sum_{s=1}^t \mathsf{KL}(f^*(a_s), f(a_s)) + \frac{1}{\rho} \ln(\delta^{-1}) \right\} \right) \leq \delta
$$

*Proof.* Using Equation ([11]) of Lemma [5], we have

$$\mathbb{P}_{f^*}\left(\exists t \geq 1, \ \sum_s^t M_s(f) - (1 + 2\sigma^2\lambda)\sum_s^t \mathbb{E}_s[M_s(f)] \geq \frac{1}{\lambda}\ln(\delta^{-1})\right) \leq \delta$$

Notice that $M_s = 2\sigma^2 \ln \frac{p_{f^*}(r_s|a_s)}{p_f(r_s|a_s)}$ and $\mathbb{E}_s[M_s(f)] = 2\sigma^2\mathsf{KL}(f^*(a_s), f(a_s))$. Then,

$$\mathbb{P}_{f^*}\left(\exists t, \ \sum_s^t \ln \frac{p_{f^*}(r_s \mid a_s)}{p_f(r_s \mid a_s)} - (1 + 2\sigma^2\lambda)\sum_{s=1}^t \mathsf{KL}(f^*(a_s), f(a_s)) \geq \frac{1}{2\sigma^2\lambda}\ln(\delta^{-1})\right) \leq \delta\ .$$

A simple change of variable concludes the proof. $\square$

Let us define $\beta_4 = \frac{2}{r^2\Lambda^2}$, $Y = \frac{w}{3}\beta_4\ln(n)$, and $A_n = \{T_2(n), T_3(n), T_4(n) \leq Y\}$ for some $w \in (0,1)$ that we tune later. Assume that $\mathbb{E}_1 T_1(n) = O(n^u)$ for some $u \in [0,1)$. Recall that we want to upper bound $\mathbb{P}_3(A_n)$ from ([21]) for which we plan to use the change of measure argument. Specifically, we observe that, for large enough $n$,

$$\mathbb{P}_1(A_n) \leq \mathbb{P}_1\left(T_1(n) \geq n - w\beta_4\ln(n)\right) \leq \frac{\mathbb{E}_1 T_1(n)}{n - w\beta_4\ln(n)} \leq \frac{\mathbb{E}_1 T_1(n)}{n/2} \overset{(a)}{\leq} c_1 n^{u-1}$$

for some constant $c_1 > 0$ where $(a)$ is by our assumption on $\mathbb{E}_1 T_1(n)$.

Because we have set $\epsilon$ to be small enough, we have $\beta_4 = \frac{2}{r^2\Lambda^2} \leq \frac{2}{\Lambda^2} \wedge \frac{2}{4\epsilon^2}$. Then, under $A_n$, we have

$$\sum_{s=1}^t \mathsf{KL}(f_3(a_s), f_1(a_s)) = T_2(n) \cdot \frac{4\epsilon^2}{2} + T_3(n) \cdot \frac{\Lambda^2}{2} + T_4(n) \cdot \frac{r^2\Lambda^2}{2}$$

$$\leq 3Y\max\left(\frac{4\epsilon^2}{2}, \frac{\Lambda^2}{2}, \frac{r^2\Lambda^2}{2}\right) \leq w\ln(n) \tag{22}$$

We now lower bound $\mathbb{P}_1(A_n)$. Recall the definition of $B(\cdot, \cdot)$ from Lemma [22].

$$\mathbb{P}_1(A_n) \geq \mathbb{P}_1\left(A_n, \overline{B(f_3, f_1)}\right)$$

$$= \mathbb{E}_3\left[\mathbb{1}\{A_n, \overline{B(f_3, f_1)}\}\prod_{t=1}^n \frac{p_1(r_t \mid a_t)}{p_3(r_t \mid a_t)}\right]$$

$$= \mathbb{E}_3\left[\mathbb{1}\{A_n, \overline{B(f_3, f_1)}\}\exp\left(-\sum_{t=1}^n \ln\frac{p_3(r_t \mid a_t)}{p_1(r_t \mid a_t)}\right)\right]$$

$$\geq \mathbb{E}_3\left[\mathbb{1}\{A_n, \overline{B(f_3, f_1)}\}\exp\left(-\left((1+\rho)\sum_{t=1}^n \mathsf{KL}(f_3(a_t), f_1(a_t)) + \frac{1}{\rho}\ln(\delta^{-1})\right)\right)\right]$$

$$\overset{(22)}{\geq} \mathbb{E}_3\left[\mathbb{1}\{A_n, \overline{B(f_3, f_1)}\}\exp\left(-\left((1+\rho)\cdot w\ln(n) + \frac{1}{\rho}\ln(\delta^{-1})\right)\right)\right]$$

$$= \mathbb{P}_3\left(A_n, \overline{B(f_3, f_1)}\right)\cdot (1/n)^{(1+\rho)w}\cdot \delta^{\frac{1}{\rho}}$$

$$\geq \left(\mathbb{P}_3(A_n) - \mathbb{P}_3(B(f_3, f_1))\right)\cdot (1/n)^{(1+\rho)w}\cdot \delta^{\frac{1}{\rho}} \qquad (\because \mathbb{P}(A) \leq \mathbb{P}(A, \overline{B}) + \mathbb{P}(B))$$

$$\geq \left(\mathbb{P}_3(A_n) - \delta\right)\cdot (1/n)^{(1+\rho)w}\cdot \delta^{\frac{1}{\rho}} \qquad (\because \text{Lemma } [22])$$

Combining the lower and upper bound on $\mathbb{P}_1(A_n)$ above,

$$\left(\mathbb{P}_3(A_n) - \delta\right)\cdot (1/n)^{(1+\rho)w}\cdot \delta^{\frac{1}{\rho}} \leq c_1 n^{u-1}$$

$$\implies \mathbb{P}_3(A_n) \leq \delta + c_1 n^{u-1}\cdot (1/n)^{-(1+\rho)w}\cdot \delta^{-\frac{1}{\rho}}$$

$$\leq (1/n)^q + c_1(1/n)^{1-u-(1+\rho)w-\frac{q}{\rho}} \qquad (\text{set } \delta = (1/n)^q)$$

By setting $w = q = \frac{1-u}{4}$ and $\rho = 1$, we have $1 - u - (1 + \rho)w - \frac{q}{\rho} = \frac{1-u}{4}$. Then, With this choice, we have

$$\mathbb{P}_3(A_n) \leq (1/n)^{\frac{1-u}{4}} + c_1 \cdot (1/n)^{\frac{1-u}{4}}$$

Using our choice of $Y_{2:4} = \frac{\beta_4}{6} \ln(n)$, we go back to where we began:

$$\frac{1}{2} \exp(-\frac{r^2 \Lambda^2}{2} \mathbb{E}_2 T_4(n))$$
$$\leq \frac{1}{\min\{Y_{2:4}\}} \left( \mathbb{E}_2 T_4(n) + \mathbb{E}_3 T_3(n) + \mathbb{E}_3 T_2(n) \right) + Q_n$$
$$\leq \frac{1}{(w/3)\beta_4 \ln(n)} \cdot 3 \cdot \left( \mathbb{E}_2 T_4(n) \vee \mathbb{E}_3 T_3(n) \vee \mathbb{E}_3 T_2(n) \right) + (1/n)^{\frac{1-u}{4}} + c_1 \cdot (1/n)^{\frac{1-u}{4}}$$

Denote by $R = \mathbb{E}_3 T_3(n) \vee \mathbb{E}_3 T_2(n) \vee \mathbb{E}_2 T_4(n)$. One can see that, if $R$ is uniformly bounded w.r.t. $n$, we get a contradiction because the LHS is bounded below but the RHS gets smaller with $n$. Therefore, $R$ must grow indefinitely over time. This implies that, for large enough $n$, we have $C \leq R$ and $(1/n)^{\frac{1-u}{4}} + c_1 \cdot (1/n)^{\frac{1-u}{4}} \leq \frac{R}{(w/3)\beta_4 \ln(n)}$. Then,

$$\frac{1}{2} \exp\left( -\frac{r^2 \Lambda^2}{2} R \right) \leq \frac{4R}{(w/3)\beta_4 \ln(n)} = \frac{(12/w)R}{\frac{2}{r^2 \Lambda^2} \ln(n)}$$

It remains to solve the above for $R$. We do so by inverting the Lambert function. Let $Z = \frac{r^2 \Lambda^2 R}{2}$. Then,

$$\exp(-Z) \leq \frac{(12/w)Z}{\ln(n)}$$
$$\frac{w}{12} \ln(n) \leq Z \exp(Z) =: X$$

We like to find $Z(X)$ that satisfies $Z(X) \exp(Z(X)) = X$. Using Orabona and Pal [34, Lemma 17], we have $\frac{3}{5} \ln(X + 1) \leq Z(X) \leq \ln(X + 1)$. Therefore,

$$Z(X) \geq \frac{3}{5} \ln(X + 1) \geq \frac{3}{5} \ln\left( \frac{w \ln(n)}{12} + 1 \right)$$

Substituting $Z(X)$ and $w$ with their definitions concludes the proof. $\qquad\square$