[Reviews · NeurIPS 2020]

Review 1

Summary and Contributions: This paper presents CROP, an algorithm for stochastic structured bandits with finite hypothesis class that is asymptotically order optimal for any structure. Differently from many existing methods, CROP does not rely on forced exploration to guarantee a minimum level of estimation accuracy but carefully chooses the hypotheses whose optimal allocations (according to different criteria) should be followed. Besided the optimality guarantees, the regret bound of CROP reveals that the algorithm (1) suffers bounded regret whenever possible and (2) does not always scale with the total number of arms.

Strengths: The algorithm presented in this paper (CROP) is novel, sound, and interesting. Its design and analysis definitely provide more insight on the general problem of deriving asymptotically optimal algorithms from problem-dependent lower bounds. I believe the latter is a relevant topic for the community and many open problems exist. This paper makes a nice contribution in this direction. Though a very recent work [1] manages to derive asymptotically optimal algorithms for general structures without requiring forced exploration, that is achieved in a more "black-box" manner, using no-regret learners with optimistic losses, while here more focus is put on the actual structure of the lower bound. In particular, I really appreaciated the efforts of the authors in understanding every component in detail. Bounded regret is also another important topic which has received little attention in works that focus on problem-dependent lower bounds. The same goes for the scaling on the number of arms. This work addresses both these problems. [1] Degenne, Rémy, Han Shao, and Wouter M. Koolen. "Structure Adaptive Algorithms for Stochastic Bandits". ICML 2020.

Weaknesses: From a theoretical perspective, CROP is proven only order optimal, while asymptotically optimal strategies (with or without forcing) exist (see also detailed comments below). Also the assumption that the set of hypothesis is finite might be quite limiting. In the supplement it is mentioned that one can deal with infinite hypothesis by using covering arguments, though that raises some computational concerns (e.g., we might have to store a prohibitive number of bandit problems). From a practical perspective, CROP seems quite computationally demanding since it requires solving different optimization problems (3 in total) for different hypotheses. Also, no experiment was provided (see detail comment below).

Correctness: All claims and methods seem correct. I quickly went through the proofs and they seem correct, though I have not provided a full detailed assessment of their validity.

Clarity: The paper is well-written, though the notation is quite heavy and not easy to track. In that regards it would be helpful to have a quick summary of the main symbols in the appendix.

Relation to Prior Work: Prior works are discusses in detail throughout the paper.

Reproducibility: Yes

Additional Feedback: 1. The notation is quite heavy and the reader is required to keep track of many terms (like the sets C(f), O(f), etc.) while going through the paper. It would be helpful to have a table of notation in the appendix. 2. Th. 1 mentions that, when \gamma^*=0, P_1=P_2=0 and the CROP suffers bounded regret. While it is clear that P_1=0 from definition, and so the main logarithmic term vanishes, it would be good to better explain why \gamma^*=0 implies P_2=0. I suppose that, if f and g are "equivalent" (and so in \mathcal{E}) their set of confusing hypotheses is also equivalent, and so if we can achieve bounded regret in f we can also achieve bounded regret in g. Is the intiution correct? 3. From a theoretical perspective, the asymptotic order-optimality could be a potential limitation since asymptotically optimal strategies exist, either with forced exploration (e.g., OAM [1] for linear bandits) or without (e.g., SPL [2] for general structures). Do you have any insight on why CROP is only asymptotically order optimal? Is it because the confidence set \mathcal{F}_t, through \beta_t, is not "tight" enough? Also, how large is the constant c_1? 4. I found interesting that the regret of CROP does not scales with the number of arms K but only with K_\Psi. However, how large/small can K_Psi be with respect to K? I suppose that K_Psi is at least the number of arms that are optimal in at least one hypothesis since the optimization problem (4) should allocate some pulls to those arms. However, if I take a linear bandit problem in which each arm is optimal for at least one hyptohesis in \mathcal{F}, don't we recover the dependence on K? 5. Regarding computational complexity, how demanding is the algorithm? It seems that CROP potentially needs to solve 3 different optimization problems (\gamma, \psi, \phi) for many different candidate hypotheses. I guess that when |\mathcal{F}| is small, it would be more efficient to precompute all those optimization problems before learning starts. Howerver, when |\mathcal{F}| is large, the computational cost (either in case of precomputation or online computation) might become prohibitive. 6. No experiments were provided. I know the main focus is theoretical but I believe some numerical results, even in very simple problems (like those used in the examples) could help in supporting the algorithm presentation. For instance, it would be interesting to see empirically how an algorithm like CROP behaves compared to those using forced exploration. Some minor comments/typos: - It would be good to provide some reference for the optimization problem in (2) - Line 122: "algoirthm" - Line 136: "provides" -> provide - Line 151: "becomes" -> become - Line 185: there is a useless bracket ")" - Line 219: period missing - The discussion of the Fallback phase (line 223) uses the definition of \phi, which is defined later in Conflict. Would it make the explanation more clear to swap the two parts? - Th. 2 in App. D: why \sigma^2 appears twice in the first two terms? [1] Hao, B., Lattimore, T. & Szepesvari, C.. (2020). Adaptive Exploration in Linear Contextual Bandit. Proceedings of the Twenty Third International Conference on Artificial Intelligence and Statistics, in PMLR 108:3536-3545 [2] Degenne, Rémy, Han Shao, and Wouter M. Koolen. "Structure Adaptive Algorithms for Stochastic Bandits". ICML 2020. UPDATE: I have read the author's rebuttal and the other reviews. I still think that the paper has merits, though the proposed approach has limitations. Moreover, I think it would be relevant to add some experiments. For these reasons, I confirm my initial view.


Review 2

Summary and Contributions: The article considers a structured variant of the multi-armed bandit problem with sub-Gaussian reward distributions, when the mean reward function f is assumed to be long to a given set F. The authors revisit structured bandits in this context by introducing an algorithm that combines optimism and pessimism, guided by lower bounds. Section 2 introduces the problem formulation, as well as key notions such as optimal regret, the oracle and several sets related to the function class. Section 3 introduces the main algorithmic contribution with details and intuition. Section 4 provides the main regret analysis (Theorem 1). Examples and discussions are further provided.

Strengths: Very well written, with clear motivations and intuitions. Strong contribution.

Weaknesses: No experiments. The algorithm and techniques seem limited to a finite function class.

Correctness: I wish I had more time to review the proofs of this article. The final result looks believable and what I checked seems ok. Given the level of polishing of the article and details given in appendix, I tend to believe the proofs are correct, but this should be checked.

Clarity: Very well written, with clear motivations and intuitions.

Relation to Prior Work: Good.

Reproducibility: Yes

Additional Feedback: I appreciate the use of the sampling in order to eliminate hypothesis from the class. That is precisely the way optimism should be understod in the first place (before the term optimism changed the intuition), and this has key implications in the context of structured arms. I also do appreciate relating each part to a simple but illustrative example, making precise how to adptively remove functions from the current class. When going beyond a finite function class, implementing the algorithm may be challenging. Also, theorem 1 becomes vacuous. What do you suggest in such cases?


Review 3

Summary and Contributions: The authors study regret minimization for the structured stochastic bandit problem. A structure is a set of hypotheses for the function that associates a mean to each arm. The paper focuses on the case of a finite hypothesis set. The authors develop an algorithm that uses a test to decide whether to exploit or explore, and that when exploring uses a new mechanism, called pessimism by the authors. Contrary to previous algorithms like OSSB, the new method does not employ forced exploration. The algorithm is proved to be asymptotically optimal up to a numerical multiplicative constant and enjoys bounded regret when possible. The regret bound does not depend on the number of arms K but on an effective number of arms K_psi, smaller than K.

Strengths: The pessimism exploration mechanism is new and the paper shows that it can lead to finite time regret bounds that lead to asymptotic optimality and allow bounded regret when that is attainable. The paper is original in its approach of the structured bandit problem. The algorithm avoids using forced exploration, which is a major source of regret for OSSB.

Weaknesses: The techniques used may depend a lot on the assumption that the hypothesis class is finite, and whether or not they could extend easily to the continuous case is not clear to me. The finiteness is not a common feature of bandit applications and the extension of the ideas developed here to the continuous case should be the reference against which the new method is evaluated. The bound depends on K_psi <= K, and the authors present an example on which K_psi << K, but my impression is that most usual structured bandit problems are such that K_psi = K (see examples further down). The computational complexity of the method appears very large, as acknowledged by the authors. There is no empirical evaluation of the proposed algorithm. That may be both a consequence of the lack of natural problem on which the number of hypotheses is finite and of the large computational complexity.

Correctness: I did not check all proofs in the appendix thoroughly.

Clarity: The paper is globally well written and explains the design of the algorithm clearly. The notations for the optimistic and pessimistic sets are very close (a tilde and a line are visually close). A possibility could be to have a line above for optimistic and a line below for pessimistic (i.e. to use \overline{} and \underline{}). Figure 1 is very helpful in understanding the different hypothesis sets.

Relation to Prior Work: Prior work is clearly discussed.

Reproducibility: Yes

Additional Feedback: It looks like in most bandit problems, gamma(f) and gamma(g) are not proportional for f != g, such that "Conflict" is entered unless the confidence set of line 10 contains only one hypothesis. Furthermore, trying to extend to the continuous case, it seems that only the "Conflict" phase of the algorithm can be triggered if the algorithm does not exploit, which is the one in which the current mechanism depends crucially on the separation between distinct hypotheses. My worry is then that this strategy will not adapt easily to the continuous case. In the appendix the authors mention the use of discretization to extend to continuous hypotheses, but then the regret term stemming from the conflict resolution could be a large constant that depends on the size of the discretization (which will probably depend on the time n). In particular, the P_2\log\log(n) term could get large. The cheating code example of page 4 has a crucial property that allows K_psi to be much smaller than K: the arms in the support of psi(f) are among the same log(K) arms for all f. The following example is convoluted due to the finite hypothesis class restriction, but it tries to mimic a linear bandit with arms in the unit sphere and hypotheses in the ball with radius approximately 10. Suppose then that we have a linear bandit problem with approximately 2^d arms corresponding to a epsilon cover of the unit sphere, and hypotheses of the form f(a) = <a,x_f> such that the hypotheses vectors are the union over a of {a}U{a + v: v orthogonal to a, among 10*2^{d-1} vectors with length in {1, ..., 10}} (i.e. up to the finiteness, that set is a disk centered at a with surface orthogonal a and the hypothesis space is the union of all those disks). Here K = 2^d and for each hypothesis f, d arms are enough to explore: the support of psi(f) should be of size d. But in that linear bandit example, for each f the vector psi(f) has different non-zero components and the quantity K_psi is still close to 2^d like K. I agree that my example is contrived, but my point is that the example of cheating codes on which K_psi<<K is also a very particular case and that I am not convinced that K_psi will be much smaller than K in problems that are closer to usual continuous cases of interest. For another, much simpler example: suppose that the hypothesis class contains all permutations of the hypotheses (i.e. if a vector (f(a))_{a\in A} is in F, where functions and vectors of values are identified, then all permutations of that vector are in F). Suppose further that there exists f such that psi(f) is not 0. Then K_psi = K. Most natural cases of structured bandits contain permutations since the structure does not mention an arm in particular, like for example lipschitz bandits or unimodal bandits. Remarks: - Line 13 of the algorithm: the last f_t of the line should be f ? - Line 227, in the definition of Delta_min, the last a^*(f) should be replaced by a.


Review 4

Summary and Contributions: The authors design new algorithms for structured stochastic bandits which can simultaneously achieve asymptotic optimality (w.r.t. log(n)) when the regret grows as O(log(n)), and achieve O(1) regret when feasible. The latter is not possible with the existing asymptotically optimal algorithms, such as OSSB, OAM, as these algorithms have \omega(1) forced exploration of suboptimal arms. The model is presented with a set of finite hypothesis, one of which (true hypothesis) maps actions to rewards. Given the past observations, it is possible to prune the set of hypotheses while retaining the true hypothesis with high probability. One standard approach is to play the best arm under the most optimistic hypothesis (a.k.a. the optimism under uncertainty principle). However, in a structured problem often such optimism fails to achieve asymptotic optimality, leading way to algorithms, such as OSSB, which use forced exploration and tracking. The ’Feasible’ mechanism in the paper uses a similar approach. The key novelty of CROP (the algorithm developed here) is eliminating forced exploration through a novel approach where pessimistic hypotheses are removed more carefully using two mechanisms, namely Fallback and Conflict. In ‘Conflict’ when multiple arms pulls are informative the tracking is done more aggressively. ‘Fallback’ is there when no arm pull is informative enough (more relaxed optimization). Finally, they present an instance ‘Cheating Code’ where the gains of CROP against Forced Sampling-based algorithms can be exponentially better in problem size.

Strengths: The paper presents a result which can achieve constant regret (if possible), and asymptotic optimality simultaneously. This result is quite novel in my view. The ideas presented in the paper are new. The authors analyze the situations why forced sampling is necessary in the asymptotically optimal algorithms. They further propose novel ways to fix that.

Weaknesses: The robustness aspect of ERM and its connection to the need for forced exploration should be described more clearly. If robust ERMs are used can we simplify the algorithm CROP? No comment on the complexity of the optimization problems 5 and 4 is made/highlighted properly (maybe I missed it). Please, comment on this for known special structured problems. The gains are only substantive when the regret is sub-logarithmic (at least asymptotically). Pointing out problems where this gain is possible will be helpful. An experiment in one of this setup comparing OSSB, OAM, and CROP will be insightful. Why does optimization 5 find an informative pull scheme when optimization 2 fails? Is it about forcing gamma_a*(f) > 0, or relaxing the set of functions for which the inequality needs to be satisfied? Edit: I have read the response, and decdied to hold my score as experimental comparisons are not promised.

Correctness: I have checked the high level proofs of the paper and they seem to be correct. I have not gone through the details.

Clarity: The paper is well written and the arguments are structured well. Some notations are hard to find (see weaknesses).

Relation to Prior Work: The paper is well placed in the literature.

Reproducibility: Yes

Additional Feedback:

[Author Response · NeurIPS 2020]

We would like to thank the reviewers for thoughtful and constructive comments. Minor suggestions and typos will be reflected in the final version, and we will cite Degenne et al, 2020.

**Computational complexity:** Reviewers have pointed out the computational complexity of the algorithm. Indeed, the computational complexity is not great and is left as future work. Our main contribution is on adapting the sample complexity to finite regret and improving the dependency on $K$ with a new algorithmic idea. More importantly, we reveal important subclasses of hypotheses in $\mathcal{F}$ that must be dealt with care, which was all hidden under forced exploration or the inflation added to the optimization problem, as researchers have done. That said, the structured bandit problem with a generic $\mathcal{F}$ covers a large variety of problems, and finding one algorithm that is optimal for all is very challenging. There are improvements to be made for popular structures like linear models; we address some of them below in response to the reviewers' comments.

**R1:** Item 2: When $P_1 = 0$, we have $\gamma^* = 0$, and those $f$'s in the equivalence class also has $\gamma^*(f) = 0$ by definition. This makes the constraint set of the optimization problem $\phi(f^*)$ empty, thus $P_2 = 0$. Item 3: The constant $c_1$ is 4. One can replace $\beta_t$ so to the order $\log(1 + t\log^2(t))$ and apply the usual analysis technique (Section 8 of Lattimore&Szepesvári, "Bandit Algorithms", 2020) to get the exact asymptotic optimality, we believe. However, this would make the analysis longer. We will make this clear in the final version. Regarding the linear case, please see our response to R3 below.

**R2:** For the confidence set for infinite hypothesis class, tools from empirical processes can help. We suggest trying out without the conflict case; we speculate that the conflict case may be automatically resolved by arm pulls made by $\psi$ for simple structures. Also, one may construct a better definition for $\psi$ as we discussed above.

**R3:** We agree that extending it to the continuous case is not immediately obvious. This is beyond the scope of the paper that we are currently working on. The algorithm may need some modifications, but the pessimism principle can still be useful, we believe. Also, the reviewer is correct that it may end up entering the conflict case a lot. However, $P_2$ may not be that large as it is related to $\mathcal{E}^*$ only, the equivalence class of $f^*$; further, many $f$'s in the disc may support the same arm. Some more detail: entering the conflict cases when $\overline{f}_t \notin \mathcal{E}^*$ will contribute to $P_3$ – these happen for a finite number of times. We believe more work is needed to deal with cases like Figure 2 while mitigating the side effect mentioned by the reviewer.

On the linear case: Thank you for the examples! While $K_\psi = O(\log(K))$ for the cheating code example, it is true that for some problems $K_\psi$ can be close to $K$ (including linear bandits as pointed out by R3). This brings up an interesting point. Clearly, our regret upper bound is far from being optimal for the example R3 provided ($K = 2^d$) since we know that the regret of OFUL does not scale like $2^d$. But then, what is the optimal finite-time instance-dependent regret here, and how does it scale with $K$? Studying finite-time lower bounds may help and is an interesting open problem.

**R4:** The optimization is often convex (e.g., linear bandits) and can be solved in a reasonable amount of time with popular software. For finite classes, they scale with the number of hypotheses, and can easily be inefficient for a very large number of hypotheses. New algorithms that do not require solving the optimization problem in the first place is an interesting open problem.

The final question refers to optimization 5, but we assume it was Eq. (4). The two issues the reviewer mentioned are indeed the reason for the change. For example, in Figure 2, imagine adding $f_5 = (.97, 1, .99, .25, .25)$. The pessimism at the beginning is $f_3$, and we will keep pulling arm A5 (the only action in the support of $\gamma(f_3)$) but $f_5$ will not be eliminated.

To answer the question on the ERM, note that the example above shows the weakness of tracking the ERM *without* forced sampling; in earlier rounds, the ERM can be $f_3$ with nontrivial probability, and it will get stuck for the same reason. We believe the ERM can still work if we properly leverage $\psi$ and $\phi$ in similar ways as CROP, which deals with docile hypotheses and conflicting hypotheses. However, we are not sure if this will make the algorithm any simpler. The forced exploration is a simple and naive solution that can replace the role of $\psi$ and $\phi$, but it brings many issues mentioned in the paper. Overall, we believe the pessimism is just a simple choice that works.

[Meta-Review · NeurIPS 2020]

Despite the lack of scalability of the proposed approach, this paper brings interesting new theoretical insights on the structured bandit problem. More comments on the complexity of the CROP algorithm and on whether doing experiments is actually possible would be appreciated.